# Unequal segregation of mitochondria during asymmetric cell division contributes to cell fate divergence in sister cells in vivo

Ioannis Segos[1,4], Jens Van Eeckhoven[1], Simon Berger [2], Nikhil Mishra[3,5], Eric J. Lambie [1] & Barbara Conradt [1] ✉

The unequal segregation of organelles has been proposed to be an intrinsic mechanism that contributes to cell fate divergence during asymmetric cell division; however, in vivo evidence is sparse. Using super-resolution microscopy, we analysed the segregation of organelles during the division of the neuroblast QL.p in *C. elegans* larvae. QL.p divides to generate a daughter that survives, QL.pa, and a daughter that dies, QL.pp. We found that mitochondria segregate unequally by density and morphology and that this is dependent on mitochondrial dynamics. Furthermore, we found that mitochondrial density in QL.pp correlates with the time it takes QL.pp to die. We propose that low mitochondrial density in QL.pp promotes the cell death fate and ensures that QL.pp dies in a highly reproducible and timely manner. Our results provide in vivo evidence that the unequal segregation of mitochondria can contribute to cell fate divergence during asymmetric cell division in a developing animal.

The generation of cellular diversity via asymmetric cell division is of fundamental importance for (nearly) all (living) organisms[1,2]. It has been proposed that differential cell fate acquisition among the daughter cells is influenced by the unequal segregation of cellular organelles, particularly mitochondria[2–4]. Strong support for this idea has been obtained in studies of budding yeast, where mitochondria are segregated unequally between the mother cell and the bud. This leads to different mitochondrial functionalities in the two cells, resulting in the rejuvenation of the bud and replicative ageing in the mother cell[5,6].

Studying mitochondrial segregation during asymmetric cell division and its impact on daughter cell fates has been more difficult in higher eukaryotic systems, in part due to technical challenges[7,8]. Unequal mitochondrial segregation has been reported in various mammalian systems, and efforts have been made to link mitochondrial segregation to differential cell fate acquisition in sister cells[9–16]. However, possibly due to cell type specificity and/or differences in experimental procedures, no common model has emerged from these studies, which were predominantly conducted in vitro. For example,

there is no consistency with respect to the role of mitochondrial dynamics (i.e., mitochondrial fusion and fission) or reactive oxygen species (ROS) in mitochondrial segregation and differential cell fate acquisition[11–15]. Importantly, none of the studies took cellular volume into account when quantifying mitochondria, and most lack real-time analyses of mitochondria as cells divide. As a result, the differences between mitochondria in sister cells and how such differences may arise during asymmetric cell division have not been clearly established in higher eukaryotic systems. In addition, whether differences between mitochondria cause or contribute to differential cell fate acquisition in sister cells in higher eukaryotic systems remains an open question.

During *C. elegans* development, 131 somatic cells undergo programmed cell death, many through apoptosis[17–20]. We previously found that these apoptotic cells are characterized by fragmented mitochondria[21]. Here, we report that this 'fragmentation' is the result of unequal mitochondrial segregation during the asymmetric cell divisions that generate cells programmed to die. Specifically, we studied the asymmetric division of the neuroblast QL.p in L1 larvae. This

[1]Research Department Cell and Developmental Biology, Division of Biosciences, University College London, London, United Kingdom. [2]Department of Molecular Life Sciences, University of Zurich, Zurich, Switzerland. [3]Faculty of Biology, Ludwig-Maximilians-University Munich, Planegg-Martinsried, Germany. [4]Present address: Helmholtz Pioneer Campus, Helmholtz Munich, Neuherberg, Germany. [5]Present address: Institute of Science and Technology Austria, Klosterneuburg, Austria. ✉e-mail: b.conradt@ucl.ac.uk

neuroblast generates a smaller daughter (QL.pp) that adopts the 'cell death' fate and undergoes apoptosis, and a larger daughter (QL.pa) that survives and divides to form two cells (QL.paa and QL.pap), which differentiate into neurons (PVM and SDQL, respectively)[17,18]. To that end, we followed mitochondrial segregation during QL.p division in real time and at super resolution. In addition to wild-type animals, we examined mutants in which either the asymmetry of QL.p division or mitochondrial dynamics are disrupted. Finally, by combining super-resolution imaging and long-term cell fate tracking, we discovered a positive correlation between mitochondrial density and the time it takes QL.pp to die. Our study represents a quantitative and real time analysis of mitochondrial segregation in an asymmetrically dividing cell in a developing animal.

## Results

We utilized two complementary approaches to follow mitochondrial segregation during QL.p (and QL.pa) division in real time and at super resolution. In one set of experiments, we used a microfluidics-based immobilization system[22] to image individual animals throughout the entire L1 stage of larval development (11–22 hours) and to acquire quantitative data on (i) mitochondrial segregation during QL.p division and (ii) mitochondrial segregation during QL.pa division (see Fig. 1a). QL.p and QL.pa were tracked using an ultrafast confocal mode, and super-resolution imaging was limited to two time points during their division (metaphase and post-cytokinesis). In another set of experiments, we used nanobeads-based immobilization[23] to image individual animals at the L1 stage for 1–2 hours and to acquire quantitative data on mitochondrial segregation during QL.p division. To determine volume, morphology and position of individual mitochondria during QL.p division, we used super-resolution imaging at high temporal resolution (1 min intervals between metaphase and post-cytokinesis). Unless noted otherwise, 3D rendering was performed and cellular and mitochondrial volumes determined using the methodology that we recently established (see Methods)[23].

### Unequal mitochondrial segregation during QL.p division

We sought to determine how mitochondria segregate during QL.p division, which generates two daughter cells that adopt different fates, and during QL.pa division, which generates two neurons (Fig. 1a, gray boxes). To quantitatively study these divisions, we used a strain in which cell membrane (myristoylated mCherry, magenta), chromatin (mCherry::histoneH1, magenta) and mitochondria (matrix-targeted GFP, cyan) are labeled specifically in Q lineage cells (*bcIs153* transgene). Imaging was performed using the microfluidics-based immobilization system (see Methods)[22].

At metaphase, the anterior (A) and posterior (P) halves of QL.p are essentially equal in volume (QL.p A/P volume ratio 1.15; Fig. 1b, top left and bottom right). Post-cytokinesis, the two daughter cells are drastically unequal in volume, with a QL.pa/QL.pp volume ratio of 3.33 (Fig. 1b, top left and bottom right). We found that mitochondria segregate unequally in terms of density (mitochondrial volume divided by cell volume) during QL.p division. Compared to the average mitochondrial density in QL.p, there is a significant increase in average mitochondrial density in the larger daughter QL.pa (from 0.041 to 0.047) and a corresponding decrease in the smaller daughter QL.pp (from 0.041 to 0.028) (Fig. 1b, top center). To correct for interindividual variance in the cellular and mitochondrial volume of the mother cell QL.p, we also determined the A/P ratios of average mitochondrial densities at metaphase (QL.p A/P) and post-cytokinesis (QL.pa/QL.pp) (Fig. 1b, top right). We found that at metaphase, the QL.p A/P mitochondrial density ratio is 0.94, indicating that mitochondria are equally distributed prior to cell division. However, post-cytokinesis, the QL.pa/QL.pp mitochondrial density ratio was 3.02, indicating unequal mitochondrial distribution among the two daughter cells irrespective of their cell size difference. This suggests that mitochondria segregate

unequally during QL.p division. To determine whether anterior-posterior mitochondrial distribution in QL.p is predictive of mitochondrial distribution in QL.pa and QL.pp, respectively, we plotted mitochondrial density ratios at metaphase against those at post-cytokinesis; however, we found no correlation (Fig. 1b, bottom left; also see Fig. S3, +/+). Therefore, mitochondrial segregation during QL.p division is not based on the distribution of mitochondria at metaphase.

In the case of QL.pa, we found that at metaphase, the anterior and posterior halves of QL.pa are essentially equal in volume with a QL.pa A/P volume ratio of 1.15; post-cytokinesis, QL.paa is slightly larger than QL.pap with a QL.paa/QL.pap volume ratio of 1.52 (Fig. 1c, top left and bottom right). We found that mitochondria are segregated equally during QL.pa division, as indicated by similar average mitochondrial densities in QL.pa and its daughter cells (0.037 vs 0.040 and 0.039), as well as similar average mitochondrial density ratios at metaphase and post-cytokinesis, which—in contrast to QL.p division—is significantly closer to 1 post-cytokinesis (1.13) (Fig. 1c, top center, top right, and bottom right). In contrast to QL.p division, we found a significant correlation between mitochondrial density ratios at metaphase (i.e., anterior-posterior mitochondrial distribution in QL.pa) and post-cytokinesis (i.e., mitochondrial distribution among the two daughter cells) (Fig. 1c, bottom left). Therefore, during QL.pa division, mitochondria segregate based on their anterior-posterior distribution at metaphase.

In summary, the segregation of mitochondria is unequal by density during QL.p division but not during QL.pa division. Moreover, the altered distribution of mitochondria after QL.p division suggests that mitochondrial segregation is a non-random actively controlled process.

### Unequal segregation may be restricted to mitochondria

Next, we wondered whether unequal segregation of organelles is a general feature of QL.p division. To answer this question, we generated strains in which cell membrane (myristoylated SFmTurquoise2ox, magenta), chromatin (SFmTurquoise2ox::his-24, magenta), mitochondria (*tomm-20*::mKate2 [in combination with lysosome labeling] or matrix-targeted GFP [in combination with ER labeling], cyan) and either lysosomes (eYFP::*cup-5*, green) (Figs. S1a and S2) or endoplasmic reticulum (ER) (SP12-mCherry-KDEL, green) (Fig. S1b) are labeled (*bcIs159* [lysosomes] and *bcIs160* [ER] transgenes). Imaging was performed at metaphase and post-cytokinesis using nanobead-based immobilization, and cellular size and organelle volumes were determined after 2D rendering (see Methods).

At metaphase, the distribution of lysosomes was slightly skewed towards the anterior side of QL.p, with an average QL.p A/P lysosome density ratio of 1.33 (Fig. S1c, left, QL.p A/P), but we observed a relatively symmetric distribution of ER (Fig. S1d, left). Post-cytokinesis, we observed essentially equal distribution of both lysosomes and ER with a subtle but significant increase in average density ratio between metaphase and post-cytokinesis in the case of ER (increase from 0.92 to 1.10) (Fig. S1c, d, left, QL.pa/QL.pp). Of note is that the average QL.pa/QL.pp mitochondrial density ratios observed using the nanobeads-based immobilization (2.02 and 1.89; Fig. S1c, d) are lower than those observed using the microfluidics-based immobilization system (3.02; Fig. 1b). However, compared to their respective mitochondrial density ratios observed at metaphase (1.15 and 1.10; Fig. S1c, d), the QL.pa/QL.pp mitochondrial density ratios are still significantly increased after division. In this set of experiments, the anterior and posterior sides of QL.p at metaphase were essentially equal in size, with QL.p A/P size ratios of 1.12 (lysosome) and 1.15 (ER). Post-cytokinesis, the two daughter cells generated were unequal in size with QL.pa/QL.pp size ratios of 3.33 (lysosomes) and 3.25 (ER) (Fig. S1c, d, right).

These results suggest that lysosomes and ER are not unequally segregated during QL.p division. Therefore, unequal segregation during QL.p division may be restricted to mitochondria.

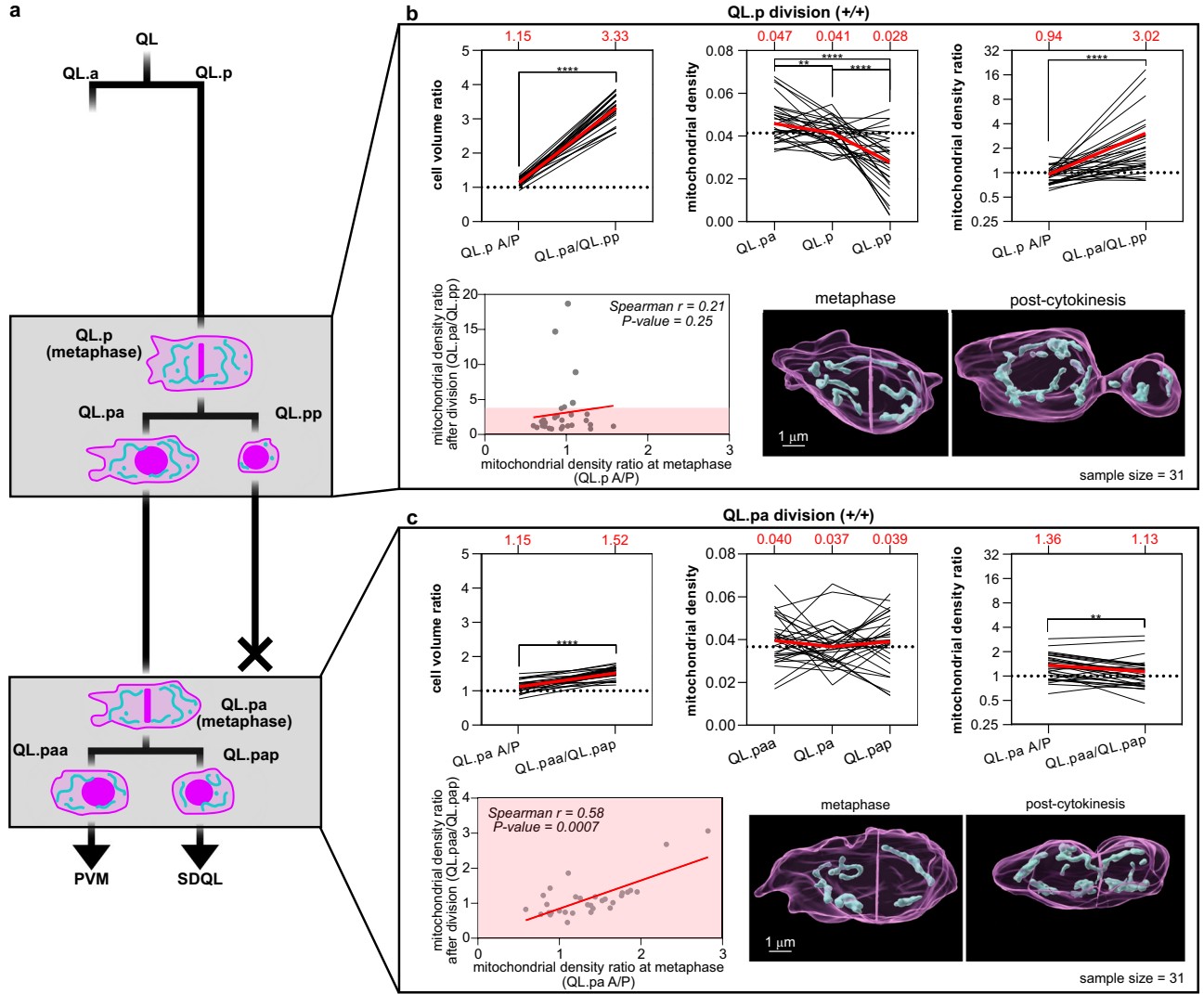

**Fig. 1 | Mitochondrial segregation is unequal and unpredictable during QL.p division, but not QL.pa division. a** Schematics of the QL.p lineage recorded in wild-type animals expressing *bcIs153* transgene. The gray boxes highlight both QL.p and QL.pa divisions. **b** Measurements of cell volume ratio, mitochondrial density and mitochondrial density ratio during QL.p division. Top left: cell volume ratio before (QL.p A/P sides) and after division (QL.pa/QL.pp); top center: mitochondrial density before (QL.p) and after division (QL.pa and QL.pp); top right: mitochondrial density ratio before (QL.p A/P sides) and aftr division (QL.pa/QL.pp); bottom left: correlation between mitochondrial density ratio before (x-axes) and after (y-axes) division. Cell volume ratio t(29) = 32.51 *p* < 0.001, QL.pa-QL.p t(30) = 3.125 *p* = 0.0017, QL.pa-QL.pp t(30) = 6.102 *p* = 0.0031, QL.p-QL.pp t(30) = 4.661 p = 0.0029, mitochondrial density ratio W = 446.0 *p* < 0.0001. **c** Measurements of cell volume ratio, mitochondrial density and mitochondrial density ratio during QL.pa division. Top left: cell volume ratio before (QL.pa A/P sides) and after division (QL.paa/QL.pap); top center: mitochondrial density before (QL.pa) and after division (QL.paa and QL.pap); top right: mitochondrial density ratio before (QL.pa A/P sides) and aftr division (QL.paa/QL.pap; bottom left: correlation between mito-chondrial density ratio before (x-axes) and after (y-axes) division; bottom right: representative 3D volumes of QL.pa division. Cell volume ratio t(30) = 10.88 *p* < 0.0001, QL.paa-QL.pa Q(31) = 0.5891 *p* = 0.3740, QL.paa-QL.pap Q(31) = 0.9439 *p* = 0.8989, QL.pa-QL.pap Q(31) = 0.5891 *p* = 0.3096, mitochondrial density ratio W = −228.0 *p* = 0.0039. *P* values are calculated using the two-sided Wilcoxon mat-ched pairs signed-rank test (both **b**, **c** top right), the paired two-sided *t*-test (both **b**, **c**, top left), the RM one-way ANOVA with the Benjamini, Krieger and Yekutieli correction (**b** top center), the Friedman test with the Benjamini, Krieger and Yekutieli correction (**c** top center), and the Spearman **c**orrelation (both **b**, **c** bottom left). Normality was tested with the Shapiro–Wilk test. *: *P* value ≤ 0.05; **: *P* value ≤ 0.01; ***: *P* value ≤ 0.001; ****: *P* value ≤ 0.0001. In the top row plots of **a**, **b**, individual black lines represent the trends of each division between metaphase and post-cytokinesis or between QL.p and QL.pa and QL.pp. Red lines and red numbers = average (both **b**, **c**, top row). *n* = 31 in **a**, **b**. Source data are provided as a Source Data file.

## Mitochondrial segregation is also unequal by morphology

To investigate the morphology and position of individual mitochon-dria during QL.p division, we used the same transgene as in Fig. 1 (*bcIs153*) in combination with nanobead-based immobilization and imaging at high temporal resolution (see Methods).

Using this experimental setup, we observed essentially the same unequal segregation of mitochondria by density during QL.p division as described above using the microfluidics-based immobilization sys-tem (compare Fig. 1b with Fig. 2a–d, +/+). We quantified mitochondrial morphology by extracting mitochondrion-specific shape parameters

and conducting a Principal Component Analysis (PCA; see Methods). While mitochondria were morphologically comparable between the anterior and posterior sides of QL.p at metaphase, they differed between QL.pa and QL.pp post-cytokinesis. On average, mitochondria in QL.pa were larger and more elongated than in QL.pp, which inher-ited mostly fragmented mitochondria. This is evidenced by the sig-nificant difference along both PC1 (variance in surface area and volume of individual mitochondria or 'size') and PC3 (variance in sphericity and surface/volume ratio of individual mitochondria or 'shape') (Fig. 2b, e, +/+).

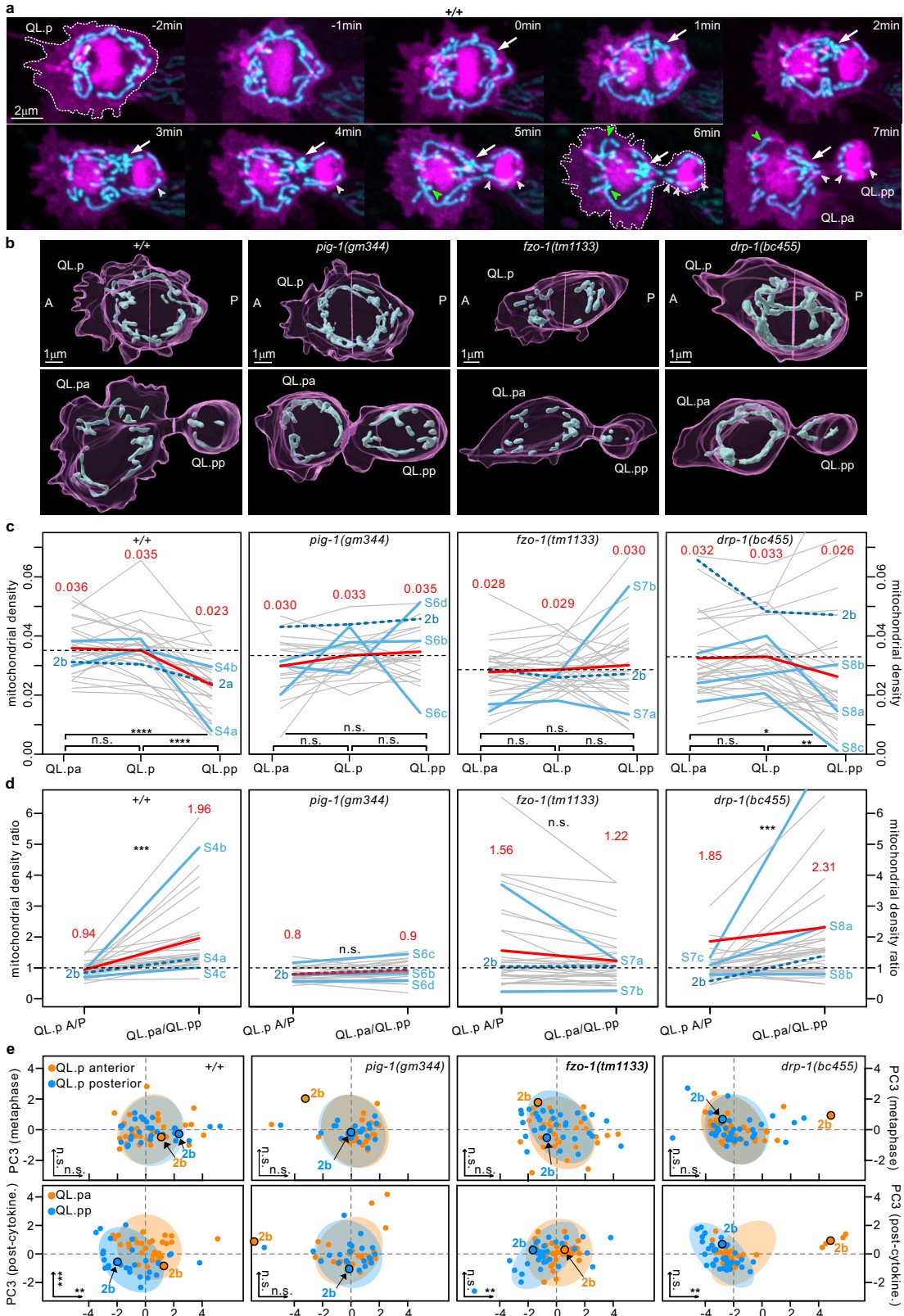

Thus, mitochondrial segregation during QL.p division is unequal not only in terms of density but also in terms of morphology.

### Role of mitochondrial transport and fission

The differences in mitochondrial density and morphology in QL.pa and QL.pp are also visually apparent in super-resolution time series (Figs. 2a and S4; compare mitochondrial morphology in QL.p at

metaphase [0 min] with mitochondrial morphologies in QL.pa and QL.pp at post-cytokinesis [7 min]). These time series also revealed that between metaphase and post-cytokinesis, mitochondria are transported from the posterior to the anterior side (Fig. 2a and S4, white arrows).

In addition, we observed mitochondrial fission and fusion events (Figs. 2a and S4, white and green arrowheads, respectively).

**Fig. 2 | Mitochondrial segregation during QL.p division is defective in *pig-1*, *fzo-1* and *drp-1* mutants. a** Super-resolution live two-color time series of QL.p division. Time series is representative of 30 divisions imaged from 30 animals. Plasma membrane (myristoylated mCherry) and chromatin (mCherry::his-24) are shown in magenta, mitochondria (mtGFP) in cyan. Images are maximum intensity projections of aligned z-stacks. In all images, anterior is left and posterior is right. Arrows point to the anteriorly directed transport of mitochondria. Green and white arrowheads point to mitochondrial fusion and fission, respectively. **b** 3D renderings of representative samples of QL.p division at metaphase (top) and post-cytokinesis (bottom) in wild type (+/+), *pig-1(gm344)*, *fzo-1(tm1133)*, and *drp-1(bc455)* (magenta: cell membrane; cyan: mitochondria). These examples refer to the blue lines in c and d and to the highlighted dots in (**e**). **c** Mitochondrial density at metaphase (QL.p) and post-cytokinesis (QL.pa and QL.pp). Blue lines refer to the 3D rendered examples in (**b**), whereas the green lines refer to time series illustrated in Figs. S4–S8. Red lines and numbers = average. Wild type (+/+): QL.p-QL.pa t(29) = 0.198, *p* = 0.593; QL.p-QL.pp t(29) = 6.934, *p* < 0.0001; QL.pa-QL.pp t(29) = 6.051, *p* < 0.0001. *pig-1(gm344)*: QL.p-QL.pa *V* = 198, *p* = 0.267; QL.p-QL.pp t(23) = 0.610, *p* = 0.548; QL.pa-QL.pp V = 220, *p* = 0.137. *Fzo-1(tm1133)*: QL.p-QL.pa t(32) = 0.801, *p* = 0.553; QL.p-QL.pp t(32) = −0.600, *p* = 0.553; QL.pa-QL.pp t(32) = 0.759, *p* = 0.553. *drp-1(bc455)*: QL.p-QL.pa V = 372, *p* = 0.120; QL.p-QL.pp *V* = 131, *p* = 0.002; QL.pa-QL.pp *V* = 138, *p* = 0.002. **d** Mitochondrial density ratio at metaphase (QL.p A-side/P-side) and post-cytokinesis (QL.pa/QL.pp). Blue lines refer to the 3D rendered examples in (**b**), whereas the green lines refer to the time series illustrated in Figs. S4–S8. Red lines and numbers represent the average. Wild type (+/+) *V* = 0,

*p* < 0.001; *pig(gm344)* t(23) = −1.678, *p* = 0.107; *fzo-1(tm1133) V* = 393, *p* = 0.044; *drp-1(bc455) V* = 114, *p* < 0.001. **e** Principal components analysis (PCA) of mitochondrial morphological parameters at metaphase (top row) and post-cytokinesis (bottom row), with confidence ellipses for principal components 1 and 3 (PC1 and PC3). Significance levels are indicated for both axes in the bottom left corner of the respective plots, with details on the analysis presented in the main text. Wild type (+/+): PC1 metaphase $F_{(1,58)}$ = 0.147, *p* = 0.703; PC3 metaphase $F_{(1,58)}$ = 0.264, *p* = 0.609; PC1 post-cytokinesis $F_{(1,58)}$ = 8.776, *p* = 0.004; PC3 post-cytokinesis $F_{(1,58)}$ = 14.463, *p* < 0.001. *pig-1(gm344)*: PC1 metaphase $F_{(1,46)}$ = 0.152, *p* = 0.699; PC3 metaphase $F_{(1,46)}$ = 0.0052, *p* = 0.943; PC1 post-cytokinesis $F_{(1,46)}$ = 0.002, *p* = 0.969; PC3 post-cytokinesis $F_{(1,46)}$ = 0.380, *p* = 0.541. *fzo-1(tm1133)*: PC1 metaphase $F_{(1,66)}$ = 0.031, *p* = 0.862; PC3 metaphase $F_{(1,66)}$ = 2.458, *p* = 0.122; PC1 post-cytokinesis $F_{(1,66)}$ = 7.756, *p* = 0.007; PC3 post-cytokinesis $F_{(1,66)}$ = 0.786, *p* = 0.379. *drp-1(bc455)*: PC1 metaphase $F_{(1,68)}$ = 0.400, *p* = 0.530; PC3 metaphase $F_{(1,68)}$ = 0.026, *p* = 0.871; PC1 post-cytokinesis $F_{(1,68)}$ = 9.869, *p* = 0.002; PC3 post-cytokinesis $F_{(1,68)}$ = 0.447, *p* = 0.506. **c** *P* values are calculated using the RM one-way ANOVA or the Friedman test with the Benjamini, Krieger and Yekutieli correction. **d** *P* values are calculated using the two-sided paired *t*-test or the two-sided Wilcoxon matched signed-rank test. Normality was tested using the Shapiro–Wilk test. **P* value ≤ 0.05; ***P* value ≤ 0.01; ****P* value ≤ 0.001; *****P* value ≤ 0.0001. All data are from animals expressing the *bcIs153* transgene, with sample sizes: wild type (+/+) *n* = 30; *pig-1(gm344) n* = 24; *fzo-1(tm1133) n* = 33; and *drp-1(bc455) n* = 35. Source data are provided as a Source Data file.

Quantifying these events revealed that fission occurs significantly more frequently in the posterior (3.17 vs 2.03 fission events) (Fig. S5a). Conversely, fusion occurs more frequently in the anterior, but not significantly so (1.37 vs 0.63 fusion events) (Fig. S5b). The ratio of fission to fusion events in the anterior is 1.48, whereas it is 5.00 in the posterior. This is consistent with balanced mitochondrial fission and fusion and the maintenance of mitochondrial morphology in the anterior, but more mitochondrial fission and, hence, mitochondrial fragmentation in the posterior. We also investigated if fission and fusion are more likely to occur during a particular stage of QL.p division. For this analysis, we defined four stages during QL.p division (Fig. S5c) and determined the number of fission and fusion events at each stage. While fission is slightly more frequent during anaphase and at the beginning of cytokinesis ('cytokinesis 1') (Fig. S5d), fusion events are more evenly distributed throughout the division (Fig. S5e). To determine whether mitochondrial fission continues in QL.pp post-cytokinesis, we used the microfluidics device to track QL.pp cells post-cytokinesis and imaged them every 40 min until their deaths (Fig. S5f). We quantified the number of mitochondria in QL.pp at all time points and found that it essentially remains constant post-cytokinesis (Fig. S5g). This suggests that post-cytokinesis, there is little mitochondrial fission (or fusion) occurring in QL.pp.

The *C. elegans* dynamin-like GTPase DRP-1 is required for mitochondrial fission and orthologous to human Drp1[24]. As mitochondrial fission occurs significantly more in the posterior half of the cell during QL.pa division, we quantified the number of DRP-1 foci, using endogenous DRP-1 protein internally tagged with mNeonGreen (DRP-1(*dx230*Internal))[25] (Fig. S5h). We found that 3D rendered DRP-1(*dx230*Internal) foci are distributed roughly symmetrically at metaphase, with marginally more DRP-1 foci occurring in the anterior (QLp A/P ratio 1.17) (Fig. S5i). Post-cytokinesis, this difference is increased with a QL.pa/QL.pp ratio of DRP-1 foci of 3.93 (Fig. S5i). Considering that the average QL.pa/QL.pp mitochondrial volume (rather than density) ratio is 6.74 (Fig. S5J), this indicates that compared to QL.pa, there are 1.72-fold more DRP-1 foci per mitochondrial volume in QL.pp. Importantly, this is similar in magnitude to the 1.56-fold increase in mitochondrial fission events observed in the posterior during QL.p division (Fig. S5a). Therefore, this increase in mitochondrial fission events may be caused by increased mitochondrial assembly of DRP-1 foci.

In summary, during QL.p division, mitochondria are transported in a targeted way, and mitochondrial fission occurs in a localized

manner. Together, these may contribute to a lower mitochondrial density and smaller, more spherical mitochondria in QL.pp.

## Mitochondrial segregation in *pig-1* MELK mutants

Next, we asked whether the loss of QL.p's ability to divide asymmetrically impacts unequal mitochondrial segregation. To address this, we examined animals homozygous for a loss-of-function (lf) mutation of the gene *pig-1* (*pig-1(gm344)*) using nanobead-based immobilization and imaging at high temporal resolution.

*pig-1* encodes a PAR-1-like kinase that is orthologous to mammalian MELK (maternal embryonic leucine zipper kinase). When *pig-1* is inactivated, QL.p divides symmetrically and gives rise to two daughter cells of equal cell volume[26] (Figs. 2b and S6a, *pig-1(gm344)*). We found that unequal mitochondrial segregation is severely affected in *pig-1*(lf) mutants: average mitochondrial densities in QL.p and its daughter cells are very similar (0.033 vs 0.030 and 0.035), and the average mitochondrial density ratios before and after division are essentially the same (0.8 and 0.9) (Fig. 2c, d). Morphological differences (PC1-size, PC3-shape) between QL.p daughter cells are lost in *pig-1*(lf) animals (Fig. 2e), and mitochondrial density ratios at metaphase correlate with mitochondrial density ratios post-cytokinesis (Fig. S3). This suggests that - unlike in wild type - mitochondria segregate based on their anterior-posterior distribution at metaphase. Finally, in super-resolution time series, we observed that in *pig-1*(lf) animals, anteriorly directed mitochondrial transport is lost and mitochondrial fission and fusion events (white and green arrowheads) occur with similar frequencies in the posterior and anterior (Fig. S5a, b and S6b–d).

Overall, in the *pig-1*(lf) background, mitochondrial segregation during QL.p division becomes equal in terms of both density and morphology.

## Mitochondrial segregation in *fzo-1* Mfn and *drp-1* Drp1 mutants

Our finding that mitochondria divide and fuse during QL.p division and the localized mitochondrial fission that occurs prompted us to investigate the role of mitochondrial dynamics in unequal mitochondrial segregation. To that end, we used nanobead-based immobilization, imaging at high temporal resolution and animals homozygous for lf mutations of either *fzo-1* (*fzo-1(tm1133)*) or *drp-1* (*drp-1(bc455)*). *fzo-1* encodes the *C. elegans* ortholog of human Mitofusins 1 and 2, which are required for mitochondrial fusion, and its loss causes mitochondrial

hyper-fission[27]. *drp-1* encodes the *C. elegans* ortholog of human Drp1, which is required for mitochondrial fission, and its loss causes mitochondrial hyper-fusion[24]. (The *drp-1* allele *bc455* was generated using CRISPR-Cas-based genome editing and is molecularly identical to the *drp-1* allele *tm1108* [see Methods]). In both mutants, QL.pa/QL.pp cell volume asymmetry is maintained (see Figs. 2b, and S7, S8).

We found that in *fzo-1*(lf) mutants, average mitochondrial densities in QL.p and its daughter cells are similar (0.029 vs 0.028 and 0.030) (Fig. 2c, *fzo-1(tm1133)*), and average mitochondrial density ratios do not change significantly between metaphase and post-cytokinesis (1.56 and 1.22) (Fig. 2d). Mitochondria appear to be distributed haphazardly at metaphase, with more mitochondria located anterior or posterior of the metaphase plate, seemingly at random (Fig. S7a–c). The mitochondrial density ratios at metaphase are also highly predictive of the density ratios post-cytokinesis (Fig. S3). Moreover, compared to wild type, we observed fewer movements of mitochondria during anaphase and cytokinesis (Fig. S7b, see arrowheads pointing to posterior mitochondria). Combined with the abnormal distribution of mitochondria at metaphase, this occasionally led to extreme outcomes, such as QL.pp inheriting either no mitochondria or much more than usual (Fig. S7c). Additionally, the morphological difference between QL.pa and QL.pp mitochondria with respect to sphericity (PC3) is lost, with only a significant difference in volume (PC1) remaining (Fig. 2e).

We observed that the loss of *drp-1* also dysregulates mitochondrial segregation, albeit in a different way. Average mitochondrial density is significantly lower in QL.pp compared to QL.p and QL.pa (0.026 vs 0.032 and 0.033) (Fig. 2c, *drp-1(bc455)*), resulting in a significant increase in average mitochondrial density ratio between metaphase and post-cytokinesis (from 1.85 to 2.31) (Fig. 2d). However, compared to wild type, the differences in mitochondrial densities and the increase in mitochondrial density ratio observed in *drp-1*(lf) animals are smaller. As in *fzo-1*(lf), the difference in sphericity of mitochondria (PC3) between the two daughter cells is lost, with only a significant difference in volume (PC1) remaining (Fig. 2e). In contrast to *fzo-1*(lf), there is no correlation between mitochondrial density ratios at metaphase and post-cytokinesis (Fig. S3). Time series of QL.p divisions reveal that mitochondria in QL.p are often hyper-fused into one large mitochondrion that encircles the aligned chromosomes at metaphase (Fig. S8a–c). When QL.p divides, QL.pp sometimes inherits a large fraction of this hyper-fused mitochondrion, whereas other times it receives only a small fraction. Consequently, mitochondrial segregation during QL.p division is not predictable.

Together, these results indicate that mitochondrial fusion and to a lesser extent mitochondrial fission contribute to the equal mitochondrial distribution in QL.p at metaphase and the unequal mitochondrial segregation during QL.p division.

To confirm these findings, we analyzed mitochondrial distribution in QL.p at metaphase and in QL.p daughter cells post-cytokinesis in *fzo-1*(lf) and *drp-1*(lf) animals using microfluidic-based immobilization. As shown in Figs. S9b and S10b, the phenotypes observed were essentially identical to those observed using nanobead-based immobilization (Fig. 2). For example, in *fzo-1*(lf) animals, no significant increase in average mitochondrial density ratio was observed between metaphase and post-cytokinesis (1.89 and 1.79) (Fig. S9b, top right) and in *drp-1*(lf) animals, compared to wild type, a significant but smaller increase was observed (increase from 1.08 to 1.48 in *drp-1*(lf) [Fig. S10b, top right] compared to an increase from 0.94 to 3.02 in +/+ [Fig. 1b, top right]).

Using microfluidics-based immobilization also allowed us to determine whether the loss of mitochondrial dynamics affects the equal segregation of mitochondria during QL.pa division. As in wild type, QL.pa division in both *fzo-1*(lf) and *drp-1*(lf) animals is essentially symmetric by cell volume (Fig. S9c top left, bottom right; Fig. S10c, top left, bottom right). In addition, we found that in both mutants, mitochondria are segregated essentially equally, as indicated by

similar average mitochondrial densities in QL.pa and its daughters QL.paa and QL.pap, as well as similar average mitochondrial density ratios before and after QL.pa division (Fig. S9c, top center and right; Fig. S10c, top center and right). As observed in wild type, we also found significant correlations between mitochondrial density ratios at metaphase and post-cytokinesis in both *fzo-1*(lf) and *drp-1*(lf) animals (Fig. S9c, bottom left; Fig. S10, bottom left). Therefore, equal mitochondrial segregation during QL.pa division is unaffected by the loss of either mitochondrial fission or fusion.

## Mitochondrial membrane potential

To determine whether there is a difference in membrane potential (Δψ) between mitochondria inherited by QL.pa and QL.pp, we used staining with tetramethylrhodamine ethyl ester (TMRE, orange) and a transgene that labels the cell membrane (myristoylated SFmTurquoise2ox, magenta), chromatin (SFmTurquoise2ox-HistoneH1, magenta) and mitochondria (mtGFP, cyan) (*bcIs158* transgene) (Fig. 3a). TMRE staining was combined with nanobead-based immobilization and 3D rendering of cell and mitochondrial volumes to measure mitochondrial membrane potential (i.e. TMRE fluorescence intensity) within individual organelles (see Methods) (Fig. 3a, b).

We determined the average TMRE fluorescence intensities of mitochondria in the anterior and posterior of QL.p at metaphase in wild type and found that they are similar (QL.p A/P ratio = 1.03) (Fig. 3c, +/+). However, post-cytokinesis, mitochondria in QL.pa showed a slightly but significantly higher TMRE fluorescence intensity than mitochondria in QL.pp (QL.pa/QL.pp ratio = 1.22) (Fig. 3c, +/+). The difference in average TMRE fluorescence intensities observed between QL.pa and QL.pp appears to suggest that mitochondria are selectively segregated based on membrane potential. However, we found that TMRE fluorescence intensity positively correlates with mitochondrial volume and negatively correlates with sphericity (Fig. S11a, b). Therefore, to correct for size- and shape-related effects, we binned individual mitochondria into six volumetric classes, where class 1 represents small and spherical mitochondria and class 6 large and tubular mitochondria (see Methods) (Figs. 3d, and S12a–c). Importantly, when we compared mitochondria of the same volumetric class, we did not find statistically significant differences in TMRE fluorescence intensities between posterior and anterior mitochondria in QL.p at metaphase or between mitochondria in QL.pa and QL.pp post-cytokinesis (Fig. 3e, metaphase, post-cytokinesis, +/+; note, each data point refers to one mitochondrion).

We also investigated the effect of altered mitochondrial dynamics on mitochondrial membrane potential during the division of QL.p (see Fig. S13). Average TMRE fluorescence intensities (as measured in QL.p cells at metaphase) are significantly lower in *fzo-1(tm1133)* and *drp-1(bc455)* mutants than in wild type (Fig. 3b, *fzo-1(tm1133)*, *drp-1(bc455)*). Furthermore, we found that the difference between metaphase and post-cytokinesis with respect to mean TMRE fluorescence ratio observed in wild type is lost in *fzo-1*(lf) and *drp-1*(lf) mutants (Fig. 3c). In both mutants, the distribution of TMRE intensities among different classes of mitochondria at metaphase is overall similar to that in wild type, with a broader range of intensities observed in *fzo-1*(lf) animals (Fig. 3e, metaphase). After division, as in wild type, class 1 mitochondria with high and low TMRE intensities are inherited by both QL.p daughters in *fzo-1*(lf) and *drp-1*(lf) mutants (Fig. 3e, post-cytokinesis). However, both mutants are characterized by a higher frequency of class 2 mitochondria in QL.pp and relatively fewer class 3–6 mitochondria in QL.pa (Fig. 3e, post-cytokinesis). These changes in the frequencies of mitochondrial classes after division likely explain the loss of asymmetry in average TMRE fluorescence ratio between QL.p daughters in these mutants.

Overall, these results suggest that the difference in average TMRE fluorescence intensities detected between QL.pa and QL.pp is not the result of biased mitochondrial segregation based on mitochondrial

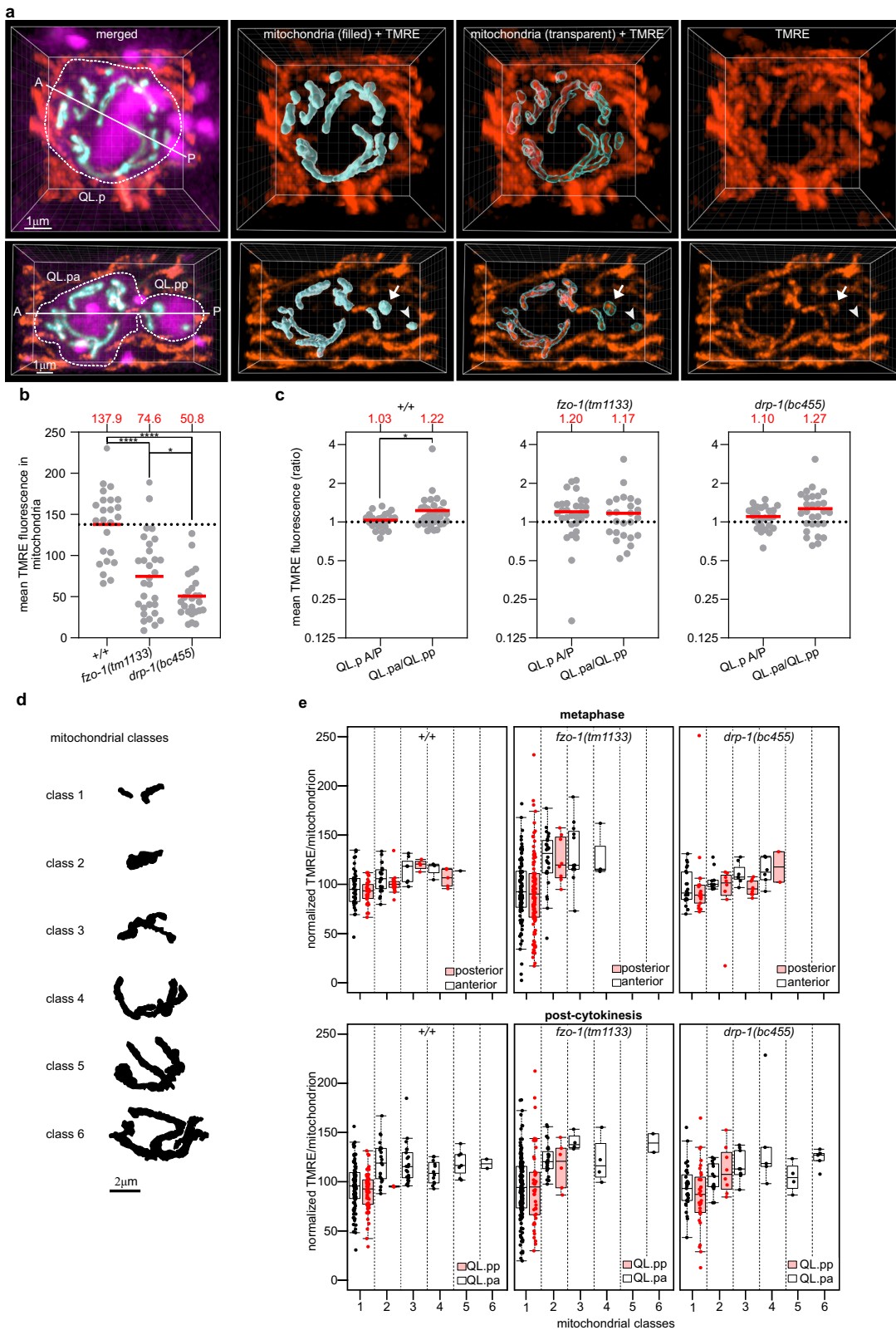

membrane potential during QL.p division. Instead, the difference in membrane potential between QL.pa and QL.pp is probably determined by the difference in mitochondrial morphologies, which are dependent on mitochondrial fusion and fission. Consequently, disrupting mitochondrial dynamics impacts mitochondrial segregation not only in terms of density and morphology, but also membrane potential.

## Mitochondrial ROS

To determine whether there is a difference in ROS between mitochondria inherited by QL.pa and QL.pp, we used animals containing two transgenes: one labeling cell membrane (myristoylated mCherry, magenta) and chromatin (mCherry-HistoneH1, magenta) (transgene *rdvIs1*)[28] and one expressing the hydrogen peroxide ($H_2O_2$) ratiometric biosensor HyPer7 targeted to the mitochondrial matrix (transgene

**Fig. 3 | Mitochondrial membrane potential does not determine unequal mitochondrial segregation during QL.p division. a** Representative 3D image of wild-type QL.p labeled with TMRE at metaphase (top row) and post-cytokinesis (bottom row). In both top and bottom left images (merged), both myristoylated SFmTurquoise2ox (cell membrane) and SFmTurquoise2ox::his-24 (chromatin) are shown in magenta, mtGFP (mitochondria) in cyan, and TMRE in orange (the same coloring is used in Fig. S11a–c). The A-P axes show the orientation of QL.p division. Arrows and arrowheads point to mitochondria with higher and lower TMRE intensities, respectively. TMRE fluorescence intensities are measured within each volume of mitochondria (see "mitochondria (3D) + TMRE" images) in Imaris. **b** Comparison of the average TMRE fluorescence intensity in wild-type (+/+) (n = 24) and mutant (fzo-1(tm1133)) (n = 30) and (drp-1(bc455)) (n = 25) animals (+/+-fzo-1(tm1133) Q < 0.0001, p < 0.0001; +/+-drp-1(bc455) Q < 0.0001, p < 0.0001; fzo-1(tm1133)-drp-1(bc455) Q = 0.0312, p = 0.0891). **c** Comparison of the mean TMRE fluorescence intensity ratio between metaphase (QL.p A/P) and post-cytokinesis (QL.pa/QL.pp) in wild type (+/+) (n = 24), fzo-1(tm1133) (n = 30), and drp-1(bc455) (n = 25). Wild type U = 273 p = 0.0471, fzo-1(tm1133) U = 332 p = 0.4756, drp-1(bc455) U = 283 p = 0.2383. **d** Outline of the six mitochondrial volume classes used in (**e**). The outlines refer to the examples given in Fig. S11. **e** Box and whisker plots of normalized TMRE fluorescence intensity per mitochondrion at metaphase (top plots) and post-cytokinesis (bottom plots) in wild type (left plots), fzo-1(tm1133) (center plots), and drp-1(bc455) (right plots). Boxes represent the interquartile range (IQR; Q1 to Q3; i.e., 50% of data) around the median (line). Whiskers represent minima and maxima, or Q1 -1.5 * IQR and Q3 + 1.5 * IQR when outliers are present. Mitochondria were sorted into six volume classes to investigate the difference in mitochondrial activity between organelles with different volumes. The mean TMRE intensity within each mitochondrion was normalized by dividing it by the mean TMRE intensity of all mitochondria in the respective cell (QL.p for metaphase or QL.pa and QL.pp, together, for post-cytokinesis) (see Methods for further information). Wild-type metaphase class1 P-class1 A mean diff. = −3.940 ± 3.274 SE (F(131) = 1.4496 p = 0.2309, class2 P-class2 A mean diff. = −5.467 ± 4.143 SE (F(131) = 1.7424 p = 0.1893, class3 P-class3 A mean diff. = 5.487 ± 8.797 (F(131) = 0.3890, SE p = 0.5339, class4 P-class4 A mean diff. = 8.371 ± 10.25 (F(131) = 0.6669) SE p = 0.4156; Wild-type post-cytokinesis (H = 23.42) class1 P-class1 A mean rank diff. = −12.58 p = 0.1427, class2 P-class2 A mean rank diff. = −56.90 p = 0.1598; fzo-1(tm1133) metaphase (H = 27.83) class1 P-class1 A mean diff. = −7.778 p = 0.4241, class2 P-class2 A mean diff. = 1.760, p = 0.9504; fzo-1(tm1133) post-cytokinesis class1 P-class1 A mean diff. = 0.8071 ± 5.418 SE (F(231) = 0.0222) p = 0.8817, class2 P-class2 A mean diff. = −5.993 ± 14.34 (F(231) = 0.1741) p = 0.6765; drp-1(bc455) metaphase (H = 20.31) class1 P-class1 A mean diff. = −8.022 p = 0.2814, class2 P-class2 A mean diff. = −10.11 p = 0.4047, class3 P-class3 A mean diff. = −23.95 p = 0.0570, class4 P-class4 A mean diff. = 2.667 p = 0.8959; drp-1(bc455) post-cytokinesis class1 P-class1 A mean diff. = −8.234 ± 6.607 (F(80) = 1.5525) p = 0.2163, class2 P-class2 A mean diff. = 7.874 ± 11.78 (F(80) = 0.4467) p = 0.5058. P values are calculated using the Kruskal–Wallis test with the Benjamini, Krieger and Yekutieli correction (**b**, **e** wild type post-cytokinesis, fzo-1(tm1133) metaphase, drp-1(bc455) metaphase), ordinary one-way ANOVA with the Benjamini, Krieger and Yekutieli correction (**e** wild type metaphase, fzo-1(tm1133) post-cytokinesis, drp1-(bc455) post-cytokinesis), and the two-sided Mann–Whitney test (**c**). Normality was tested with the Shapiro–Wilk test. *: P value ≤ 0.05; **: P value ≤ 0.01; ***: P value ≤ 0.001; ****: P value ≤ 0.0001. **b**, **c** red lines and dots represent the population average and the individual samples' average, respectively. **e** Dots represent individual mitochondria. All data are from animals expressing the bcIs158 transgene. Source data are provided as a Source Data file.

bcEx1422)[29]. ROS in the form of superoxide ($O_2^-$) produced by the electron transport chain (ETC) converts into $H_2O_2$ through dismutation; hence, levels of $H_2O_2$ determined using HyPer7 reflect mitochondrial ROS levels. Since the HyPer7 sensor is expressed under the control of a ubiquitously active promoter (Fig. S14a), we removed fluorescence intensities outside QL.p or QL.pa and QL.pp using Fiji and the Q lineage-specific transgene rdvIs1 (Fig. S14b, see Methods). Next, using Imaris, we measured mean HyPer7 fluorescence intensities within individual mitochondria excited at wavelengths of 405 nm and 488 nm (Fig. S14b).

We determined the average HyPer7 ratiometric fluorescence intensities (488 nm/405 nm) of mitochondria in the anterior and posterior of QL.p at metaphase and found that their ratio is 1.03 (Fig. S14c). Similarly, the ratio of the average HyPer7 ratiometric fluorescence intensities of mitochondria in QL.pa and QL.pp post-cytokinesis is 1.10 (Fig. S14c). To correct for size- and shape-related effects, we binned individual mitochondria into six volumetric classes, where class 1 represents small and spherical mitochondria and class 6 large and tubular mitochondria (see Methods) (Fig. 3d). When we compared mitochondria of the same volumetric class, we did not find statistically significant differences in HyPer7 ratiometric fluorescence intensities between posterior and anterior mitochondria in QL.p at metaphase or between mitochondria in QL.pa and QL.pp post-cytokinesis (Fig. S14d, metaphase, post-cytokinesis; note, each data point refers to one mitochondrion). (However, we found that the average HyPer7 ratiometric values for nearly all volumetric classes increased from below 1 at metaphase to above 1 post-cytokinesis. This suggests that the intracellular redox state and consequently mitochondrial redox state of QL.p changes during its division [Fig. S14d]). Finally, we found that HyPer7 ratiometric fluorescence intensities correlate with mitochondrial volume, but not sphericity, post-cytokinesis. However, it correlated with neither mitochondrial volume nor sphericity at metaphase (Fig. S14e, f).

In summary, these results demonstrate that there is no difference in ROS levels and, hence, redox states between mitochondria in QL.pa and QL.pp. Therefore, ROS and redox state are unlikely to determine unequal mitochondrial segregation during QL.p division.

## Unequal mitochondrial segregation and daughter cell fates

Since QL.p distributes mitochondria unequally between the two daughter cells and the two daughter cells acquire distinct cell fates, we investigated whether unequal mitochondrial segregation correlates with the divergence of these fates. For this experiment, we used microfluidic-based immobilization for long-term tracking of Q lineage cells and acquired quantitative data on (i) QL.p cell cycle length (time between QL division and QL.p division), (ii) QL.pa cell cycle length (time between QL.p division and QL.pa division) as a measure of QL.pa fate, (iii) QL.pp survival time (time between QL.p division and QL.pp death; Fig. S15, see Methods for description of how QL.pp survival time was determined) as a measure of QL.pp fate, (iv) mitochondrial segregation during QL.p division and (v) mitochondrial segregation during QL.pa division (see Fig. 4a). QL.p and its daughter cells were tracked using an ultrafast confocal mode. Super-resolution imaging during QL.p and QL.pa division was performed at metaphase and post-cytokinesis (see Methods).

We found that in wild-type animals, the cell cycle lengths of QL.pa and QL.p strongly correlate (Fig. 4b), while QL.pp survival time does not correlate with QL.p cell cycle length (Fig. 4c). Furthermore, average mitochondrial density ratio does not correlate with QL.pa cell cycle length (Fig. 4b), yet it shows a negative but non-significant correlation with QL.pp survival time (Fig. 4c). This suggests that QL.p and QL.pa cell cycles may progress at a rate proportionate to the overall developmental rate of the animal. In contrast, QL.pp survival time is independent of developmental rate, but may negatively correlate with the level of asymmetry in mitochondrial segregation. In other words, a smaller mitochondrial density ratio (i.e. QL.pp inherits relatively more mitochondria) may correlate with longer QL.pp survival times. Conversely, a larger mitochondrial density ratio (i.e., QL.pp inherits relatively fewer mitochondria) may correlate with shorter QL.pp survival times.

Next, we analyzed fzo-1(tm1133) and drp-1(bc455) animals, in which unequal mitochondrial segregation is compromised and more variable. As in wild type, QL.pa cell cycle length strongly correlates with QL.p cell cycle length in both fzo-1(lf) and drp-1(lf) animals, but mitochondrial density ratio does not correlate with QL.pa cell cycle length

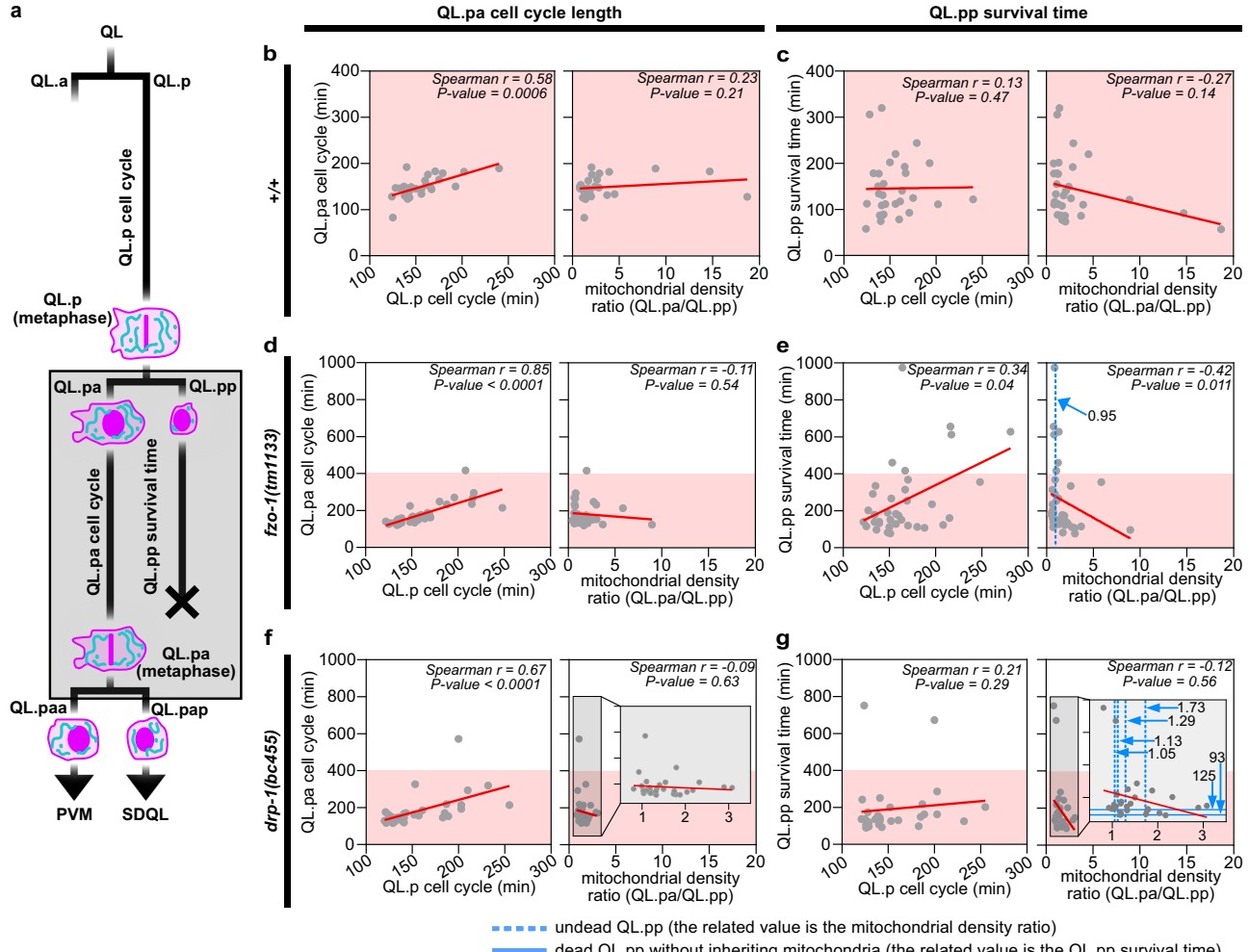

**Fig. 4 | Analysis of the relationship between mitochondrial segregation and QL.pa and QL.pp fates. a** Schematics of the QL.p lineage recorded in wild type (+/+), *fzo-1(tm1133)* and *drp-1(bc455)* animals expressing the *bcIs153* transgene. The gray box indicates the time QL.pa and QL.pp cell fates were observed, which was done in super resolution (metaphase and post-cytokinesis time points for QL.pa) and in high resolution (morphological changes [i.e., cell death] for QL.pp [see Fig. S10]), respectively. QL.p division was also followed in super resolution at metaphase and post-cytokinesis. **b, c** Correlation between QL.pa cell cycle length (**b**) or QL.pp survival time (**c**) and QL.p cell cycle length (left) or mitochondrial density ratio (right) in wild type (+/+). **d, e** Correlation between QL.pa cell cycle length (**b**) or

QL.pp survival time (**c**) and QL.p cell cycle length (left) or mitochondrial density ratio (right) in *fzo-1(tm1133)*. **f, g** Correlation between QL.pa cell cycle length (**b**) or QL.pp survival time (**c**) and QL.p cell cycle length (left) or mitochondrial density ratio (right) in *drp-1(bc455)*. Red lines = linear regression fitted on scattered plots. Dots represent data from single QL.p divisions. All correlations were analyzed calculating the Spearman correlation coefficient *r* and its related *p* value. *n* = 31, 35, and 33 in wild type (+/+), *fzo-1(tm1133)* and *drp-1(bc455)*, respectively. Data were collected from animals expressing *bcIs153* transgene. Source data are provided as a Source Data file.

(Fig. 4d, f). In *fzo-1*(lf), the survival time of QL.pp shows a weak positive correlation with QL.p cell cycle length (Fig. 4e), but this is not the case for *drp-1*(lf) (Fig. 4g). In addition, in *fzo-1*(lf), average mitochondrial density ratio shows a significant negative correlation with QL.p survival time (Fig. 4e). In *drp-1*(lf), as in wild type, average mitochondrial density ratio shows a negative but non-significant correlation with QL.pp survival time (Fig. 4g). Importantly, in both *fzo-1*(lf) and *drp-1*(lf) mutants, we captured QL.p lineages in which QL.pp inappropriately survived. In the *fzo-1*(lf) background, we captured one QL.p lineage in which QL.pp survived. This particular QL.pp cell inherited relatively more mitochondria with an average QL.pa/QL.pp mitochondrial density ratio of 0.95 rather than the average 1.74 (Fig. 4e, dotted vertical blue line). In the *drp-1*(lf) background, we captured four QL.p lineages in which QL.pp survived. Three of these QL.pp cells inherited relatively more mitochondria with average QL.pa/QL.pp mitochondrial density ratios between 1.05 and 1.29 rather than the average 1.48 (Fig. 4g, dotted vertical blue lines). The fourth QL.pp cell inherited slightly fewer mitochondria with an average QL.pa/QL.pp mitochondrial

density ratio of 1.78. In the *drp-1*(lf) background, we also captured two lineages, in which QL.pp did not inherit any mitochondria. In both cases, QL.pp died markedly faster with a survival time of only 93 min or 125 min, compared to an average survival time of 208 min in *drp-1*(lf) (Fig. 4g, horizontal blue lines and Fig. S16, left).

In order to be able to include the cases of inappropriately surviving QL.pp cells in *fzo-1(tm1133)* and *drp-1(bc455)* animals in our analyses, we used the Kaplan–Meier estimate[30]. QL.pp survival probabilities of the different genotypes are shown in the Kaplan–Meier survival probability plot (Fig. 5a). The loss of *fzo-1* or *drp-1* shows a significant effect on QL.pp survival probability compared to wild type (+/+) (Fig. 5a; *p* = 0.0044). In wild-type animals, essentially all QL.pp cells die within 300 min post-cytokinesis. In contrast, in *fzo-1(tm1133)*, some QL.pp cells die only after 900 min, and in *drp-1(tm1108)*, some survive longer than 1200 min. While a non-parametric and univariate Kaplan–Meier analysis is robust, it provides little information beyond the effect of genotype. In survival analysis, the multivariate Cox regression[31] can be used when there are multiple potentially

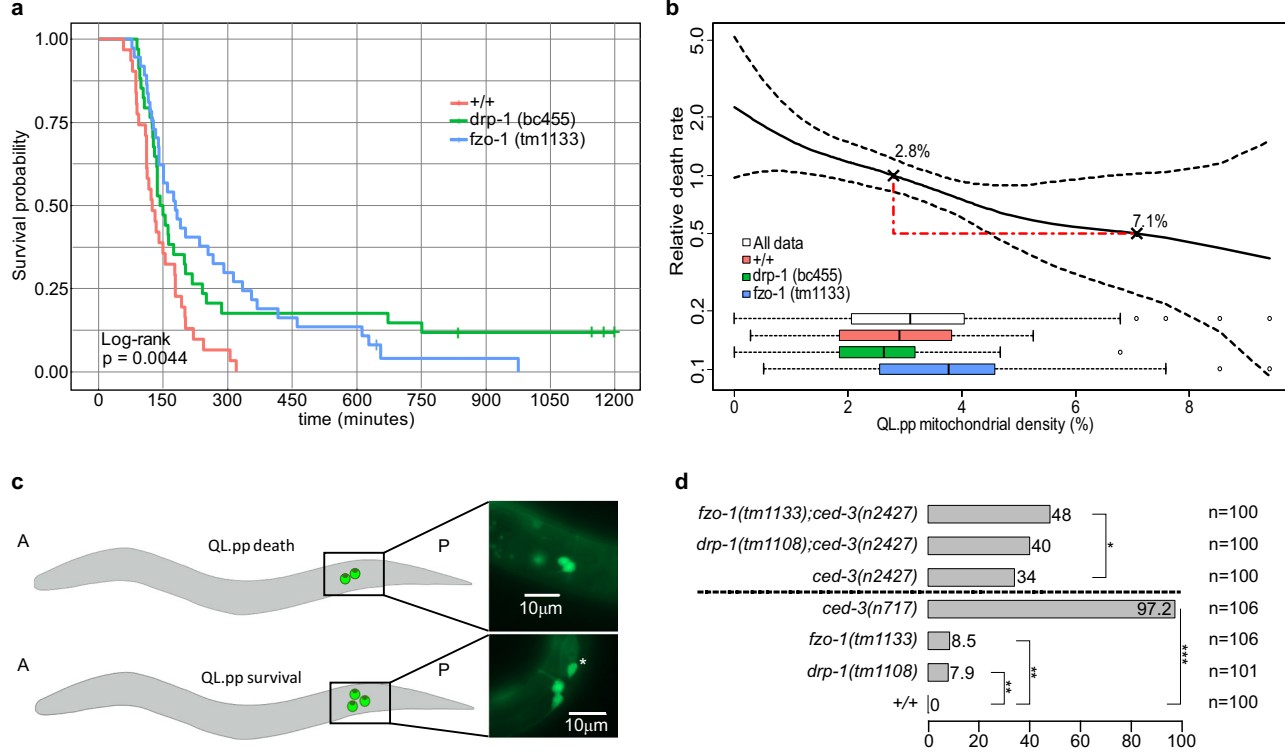

**Fig. 5 | Impact of mitochondrial density and mitochondrial dynamics on QL.pp cell death fate. a** Kaplan–Meier survival curves for wild type (+/+) (n = 31; RMST = 145 min ±11.4SE), *fzo-1(tm1133)* (n = 37; RMST = 261 min ±31.0SE, and *drp-1(bc455)* (n = 34; RMST = 254 min ±39.3se), with increased survival in loss of function mutants ($X^2$(2) = 10.9, p = 0.004). Ticks along the curves represent QL.pp cells that were phenotypically determined to survive and subsequently no longer followed (right censored data; upper limit RMST set to 751 min, i.e., earliest surviving QL.pp cell). Global log-rank test indicates significant differences in survival between the genotypes. **b** Relative death rate explainable by the mitochondrial density in QL.pp, as obtained by the Cox proportional-hazards regression (*fzo-1(tm1133)* LH = −0.489 ± 0.265SE, p = 0.065; *drp-1(bc455)* LH = −0.757 ± 0.271SE, p = 0.005; mitochondrial density LH = −0.196 ± 0.064SE, p = 0.002). Dotted lines indicate pointwise standard errors. The spread of the data across different genotypes is illustrated by the inset boxplots (+/+ n = 31, *fzo-1(tm1133)* n = 37, *drp-1(bc455)* n = 34). Boxes represent the interquartile range (IQR; Q1 to Q3; i.e., 50% of data) around the median (line). Whiskers represent minima and maxima, or Q1−1.5 * IQR and Q3 + 1.5

* IQR when outliers are present. Death rates are relative to the average mitochondrial density in QL.pp (2.8%; Fig. 1b), with the example showing a halving of death rate in QL.pp at a mitochondrial density of 7.1%. **c** Schematics of the QL.pp survival assay. Cell counting is conducted in early L2 larvae, where only two cells (top) are visible (QL.paa (PVM) and QL.pap (SDQL)) in wild-type animals. Whenever QL.pp survives, a third cell is counted (asterisk, bottom), or even four cells can be counted (divided QL.pp, example not shown). **d** QL.pp survival comparisons of wild type (+/+) with: *drp-1(tm1108)* (one-sided p = 0.006), *fzo-1(tm1133)* (one-sided p = 0.005), and *ced-3(n717)* (one-sided p < 0.001) (lower side). And QL.pp survival comparisons of *ced-3(n2427)* with: *fzo-1(tm1133); ced-3(n2427)* (one-sided p = 0.038) and *drp-1(tm1108); ced-3(n2427)* (one-sided p = 0.232) (upper side). P values refer to Fisher's Exact tests:. *: P value ≤ 0.05; **: P value ≤ 0.01; ***: P value ≤ 0.001; ****: P value ≤ 0.0001. Data in **a**–**d** were derived from animals expressing the transgenes *bcIs153* or *bcIs133*, respectively. **a**, **b** n = 31, 36 or 37 in wild type (+/+), *fzo-1(tm1133)* or *drp-1(bc455)*, respectively. RMST restricted mean survival time, LH log hazard. Source data are provided as a Source Data file.

interacting covariates, such as for instance genotype and mitochondrial density. Using this proportional hazard model, we can relate the time before QL.pp death occurs (QL.pp survival time) to our covariates through the hazard function. This allows us to estimate the effect of mitochondrial density, while still accounting for the confounding effect of genotype in our time-to-event survival data. Holding the effect of genotype constant, we found that a 1% percent increase in mitochondrial density increases QL.pp's life expectancy by 17.8% (p = 0.002). Looking at the relative death rate (which is cumulative over the entire experiment) (Fig. 5b), we see that increasing QL.pp mitochondrial density from 2.8% (average for wild type) to 7.1% leads to a halving of the relative death rate (from 1.0 to 0.5) (Fig. 5b). Furthermore, once mitochondrial density is accounted for in the Cox regression model, the loss of *fzo-1* no longer shows a significant effect on QL.pp survival (p = 0.065). This suggests that the loss of *fzo-1* affects QL.pp survival exclusively through defects in mitochondrial segregation. However, the loss of *drp-1* maintains its significant effect on survival (p = 0.005). This suggests that the effect of *drp-1*(lf) on QL.pp survival is not solely due to *drp-1*'s contribution to mitochondrial

segregation. Hence, the loss of *drp-1*(lf) affects QL.pp survival through a second, mitochondrial segregation-independent process.

Finally, we verified the findings obtained from the microfluidic experiment and the analyses of relative death rate and survival probability of QL.pp by conducting a QL.pp survival assay using transgenic animals that express a Q lineage-specific GFP transgene (*bcIs133* transgene)[32] (Fig. 5c). In wild type, QL.pp always dies (0% inappropriate survival) (Fig. 5d, +/+), whereas in control animals lacking the caspase *ced-3*, 97.2% of QL.pp cells inappropriate survive (Fig. 5d, *ced-3(n717)*). We found that both *fzo-1(tm1133)* and *drp-1(tm1108)* animals exhibited low, but significant, inappropriate QL.pp survival compared to control animals (8.5 and 7.9%, respectively). In the background of the weak *ced-3* loss-of-function mutation *n2427*, neither *fzo-1(tm1133)* nor *drp-1(tm1108)* showed a synergistic effect in terms of inappropriate QL.pp survival (Fig. 5d).

In conclusion, our findings are consistent with the notion that unequal mitochondrial segregation occurs during QL.p division and, hence mitochondrial density in QL.pp, correlates with the acquisition of the cell death fate by QL.pp. Specifically, low mitochondrial density

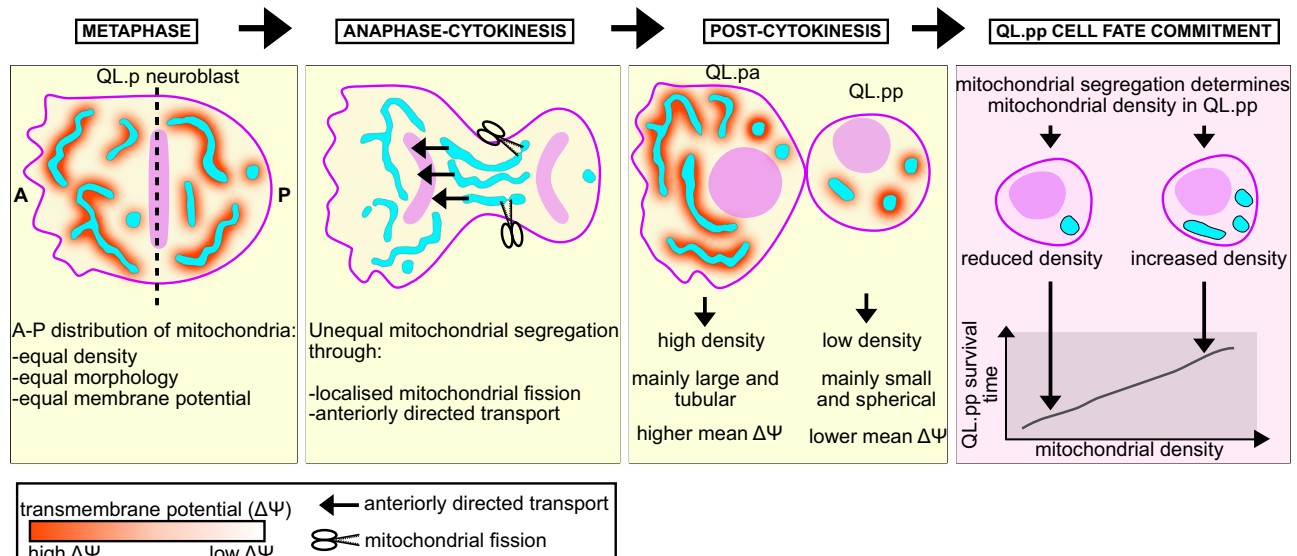

**Fig. 6 | Working model of unequal mitochondrial segregation during QL.p division.** Mitochondria are symmetrically distributed at metaphase in terms of density, morphology, and membrane potential. This symmetric distribution is dependent on mitochondrial dynamics (Metaphase). Localized mitochondrial fission and anteriorly directed mitochondrial transport contribute to unequal mitochondrial segregation during QL.p division (Anaphase-Cytokinesis). In contrast, mitochondrial segregation during QL.p division appears stochastic with respect to mitochondrial membrane potential. As a result, the surviving daughter QL.pa inherits more and larger mitochondria than QL.pp, which is programmed to die (Post-cytokinesis). Mitochondrial density in QL.pp–as determined by unequal mitochondrial segregation during QL.p division – affects QL.pp survival time and, hence, the ability of QL.pp to die (QL.pp cell fate commitment). See Discussion for a more detailed description.

in QL.pp is associated with very rapid QL.pp death, whereas high density coincides with inappropriate QL.pp survival.

## Discussion

### Mitochondria segregate unequally during QL.p division

Unequal mitochondrial segregation has been demonstrated in budding yeast but has been difficult to study in higher eukaryotic systems. Taking advantage of the highly reproducible development of *C. elegans* and advances in live-cell super-resolution microscopy, we have developed a methodology that enables us to quantitatively study mitochondrial segregation in real time and at single organelle resolution during asymmetric cell divisions in *C. elegans* larvae. Using this methodology, we have obtained highly reproducible data, even when using different transgenes, immobilization methods, growth media and bacterial foods. These data demonstrate that mitochondrial segregation during the asymmetric division of the neuroblast QL.p is unequal by density and morphology. Consequently, compared to its sister cell (QL.pa), QL.pp has a lower density of mitochondria, and its mitochondria are predominantly small and spherical. Therefore, unequal mitochondrial segregation during asymmetric cell division also occurs in a higher eukaryotic system, i.e., an animal system. We also detected lower membrane potential per mitochondrial volume in QL.pp; however, our data suggest that this is due to the overrepresentation of small and spherical mitochondria in QL.pp. In contrast to mitochondrial membrane potential, we did not detect any difference between QL.pa and QL.pp in terms of mitochondrial ROS levels and redox states, which is only weakly influenced by mitochondrial morphology.

We previously showed that cells programmed to die during *C. elegans* development are characterized by fragmented, i.e., small and spherical mitochondria[21]. Ding Xue and colleagues proposed that this is the result of the general dismantling of cells undergoing caspase-dependent apoptosis[33]. We now present evidence that this mitochondrial fragmentation (i.e., mitochondrial fission) does not occur post-cytokinesis. Instead, it is the result of unequal mitochondrial segregation during the asymmetric cell divisions that generate cells programmed to die.

### Mitochondrial segregation during QL.p division is non-random

We propose that mitochondrial segregation during QL.p division is an actively controlled process and therefore non-random. First, we found that during symmetric QL.pa division, mitochondrial distribution in the daughter cells (QL.paa and QL.pap) is predictable based on mitochondrial distribution in QL.pa at metaphase. This suggests that during QL.pa division, mitochondria segregate based on their distribution at metaphase. Since mitochondria are equally distributed throughout the anterior and posterior halves of QL.pa, mitochondria are equally segregated into QL.paa and QL.pap. In contrast, during asymmetric QL.p division, mitochondrial distribution in the daughter cells (QL.pa and QL.pp) is not predictable based on mitochondrial distribution in QL.p at metaphase. Since mitochondria are also equally distributed throughout the anterior and posterior halves of QL.p, mechanisms must be at work to control mitochondrial segregation and cause unequal mitochondrial distribution between QL.pa and QL.pp. Indeed, we present evidence that these mechanisms include anteriorly directed mitochondrial transport and localized mitochondrial fission (see below, Fig. 6). Second, the loss of mitochondrial fusion (i.e., *fzo-1(tm1133)* background) makes mitochondrial distribution in QL.pa and QL.pp predictable based on mitochondrial distribution in QL.p. Importantly, daughter cell size asymmetry is maintained in *fzo-1(tm1133)* mutants (Fig. S9). Therefore, the processes that establish daughter cell size asymmetry are not sufficient to cause unequal mitochondrial distribution in QL.pa and QL.pp, at least in the *fzo-1(tm1133)* background. This argues against a model in which unequal mitochondrial segregation is caused by daughter cell size asymmetry. Interestingly, mammalian Mitofusin 2 has been shown to interact with the Miro/Milton complex and to be required for mitochondrial transport in axons[34]. Therefore, FZO-1 may directly be involved in anteriorly directed mitochondrial transport during QL.p division. Finally, mitochondrial distribution in QL.pa and QL.pp also becomes predictable in *pig-1(gm344)* animals, and this may at least in part be due to loss of anteriorly directed mitochondrial transport and localized mitochondrial fission. The PAR-1-like kinase PIG-1 MELK is required for QL.p polarization along the anterior-posterior axis and the establishment of

daughter cell size asymmetry during QL.p division. We found that anteriorly directed mitochondrial transport, localized mitochondrial fission and the establishment of daughter cell size asymmetry occur simultaneously during QL.p division (see Figs. 2a and S4). Therefore, we consider it likely that anteriorly directed mitochondrial transport and localized mitochondrial fission are lost in *pig-1*(lf) mutants as a result of the loss of *pig-1*'s role in QL.p polarization, rather than its role in the establishment of daughter cell size asymmetry. Interestingly, *Xenopus* MELK has been shown to localize to mitochondria[35]. It is, therefore, conceivable that PIG-1 may also play a direct role in mitochondrial segregation, for example, in anteriorly directed mitochondrial transport.

We propose the following model for how mitochondria are unequally segregated during asymmetric QL.p division (Fig. 6). At metaphase, mitochondria are equally distributed throughout the anterior and posterior halves of QL.p with respect to density, morphology and membrane potential. This equal distribution is dependent on the ability of mitochondria to divide and fuse and, hence, most likely on 'normal' mitochondrial morphology. During QL.p division, small and spherical mitochondria become enriched in the posterior half, which will form the smaller daughter QL.pp; this is dependent on at least two processes, both of which are dependent on PIG-1 MELK: (i) mitochondrial fission in the posterior half and (ii) anteriorly directed transport of larger and more tubular mitochondria. What could activate mitochondrial fission in the posterior half of QL.p as it divides? We previously showed that a low level of CED-3 caspase is activated in QL.p and that prior to QL.p division, an anteroposterior gradient of this CED-3 caspase activity is established in QL.p with higher levels of CED-3 caspase in the posterior of QL.p[32]. Furthermore, Ding Xue and colleagues reported that DRP-1 can be activated through CED-3 caspase-dependent cleavage[33]. Hence, mitochondrial fission could be activated in the posterior of QL.p through localized CED-3-dependent DRP-1 cleavage. Our finding that there are more DRP-1 foci per mitochondrial volume on the posterior suggests that CED-3-dependent DRP-1 activation may promote DRP-1 foci assembly on the outer mitochondrial membrane. Concerning anteriorly directed transport of mitochondria, we propose that it is mediated by cytoskeletal elements and specific motor proteins. Our results also suggest that neither small spherical mitochondria nor large hyper-fused mitochondria are competent for anteriorly directed transport. Therefore, we propose that anteriorly directed transport is selective based on mitochondrial morphology. Finally, because we observed no difference in mitochondrial membrane potential or ROS levels and redox state when comparing mitochondria of similar volumes and sphericities in QL.pa and QL.pp post-cytokinesis, our results do not support the notion that these mitochondrial attributes contribute to selective mitochondrial transport.

## Unequal mitochondrial segregation contributes to daughter cell fates

As a result of unequal mitochondrial segregation during QL.p division, compared to its sister cell (QL.pa), QL.pp has a lower density of mitochondria. We propose that this lower density of mitochondria in QL.pp is required for the timely and reproducible death of QL.pp and, hence, the divergence of the QL.pa and QL.pp fates during asymmetric QL.p division. Using three different genetic backgrounds (wild type, *fzo-1*(lf), *drp-1*(lf)), we consistently found that changes in mitochondrial density in QL.pp impact the ability of QL.pp to adopt the cell death fate. Specifically, we found that decreased mitochondrial density in QL.pp correlates with QL.pp dying faster. Conversely, increased mitochondrial density in QL.pp correlates with QL.pp dying more slowly or not at all. It remains to be determined whether changes in mitochondrial density are sufficient to cause changes in QL.pp survival time. However, our findings are consistent with the idea that mitochondrial density impacts the activation of the apoptosis pathway and/or the execution of apoptotic cell death in QL.pp. Of note, the

accompanying changes in mitochondrial densities in QL.pa had no impact on QL.pa cell cycle time. However, as a result of QL.pa's size (~3 times the size of QL.pp), these changes may have been too subtle to have an impact. Consistent with this, in wild-type animals (microfluidics-based immobilization), mitochondrial densities in QL.pa range from 0.033 to 0.068 units and in QL.pp from 0.003 to 0.053 units.

Cell-to-cell variability in mitochondrial volume has previously been proposed to contribute to cellular noise and differences, for example, in transcriptional rates in mammalian cells[36,37]. Furthermore, mitochondrial volume or density has also been proposed to impact the activation of apoptosis in mammalian cells. However, while one study found that mitochondrial density positively correlates with resistance to TRAIL-induced apoptosis[38] (consistent with our findings in *C. elegans*), another study found that cells with increased mitochondrial volume are more prone to die[39]. One possible reason for the discrepancy between these studies could be the fact that cellular volume was not taken into account in the second study.

Why might increasing mitochondrial density in QL.pp cause QL.pp to die more slowly or not at all? The anti-apoptotic Bcl-2 protein CED-9 localizes to the outer mitochondrial membrane[40]. Therefore, CED-9 levels are expected to increase in QL.pp with increasing mitochondrial density. Furthermore, CED-9 is a substrate of CED-3 caspase[41], which we have confirmed in vitro[42]. Therefore, it is conceivable that increased CED-9 levels in QL.pp compete with other CED-3 caspase substrates, thereby compromising the ability of CED-3 caspase to trigger the execution of apoptotic cell death. Alternatively, apoptotic cell death might be compromised by other mitochondrial proteins or mitochondrial metabolites, which are also expected to increase in QL.pp with increasing mitochondrial density. Changes in levels of anti-apoptotic CED-9, other mitochondrial proteins or mitochondrial metabolites may also explain why QL.pp dies faster when mitochondrial density is decreased or – in some cases – no mitochondria are inherited by QL.pp. How the apoptosis pathway is fully activated in QL.pp in the absence of mitochondria is an enigma, unless it is triggered and the CED-4 Apaf1 apoptosome is assembled prior to the completion of QL.p division. We consider it unlikely that the low level of CED-3 caspase activity that is specifically inherited by QL.pp as a result of the formation of a CED-3 caspase gradient in QL.p (see above) is sufficient to kill QL.pp, even less so to kill QL.pp faster than at normal mitochondrial density.

Finally, our data indicate that, in contrast to *fzo-1*, *drp-1*'s impact on QL.pp survival is not limited to its role in unequal mitochondrial segregation during QL.p division and, hence, its role in mitochondrial fission. Interestingly, we previously showed that DRP-1 protein - but not FZO-1 protein - can interact with CED-9 Bcl-2 bound to the pro-apoptotic BH3-only protein EGL-1[43]. Therefore, we posit that *drp-1* may impact QL.pp survival also through a DRP-1/CED-9/EGL-1 complex, which may directly affect the activation or activity of the apoptosis machinery in QL.pp. Through what mechanism the DRP-1/CED-9/EGL-1 complex may affect apoptosis remains to be determined.

In conclusion, we provide evidence that the unequal segregation of mitochondria can contribute to cell fate divergence in sister cells in a developing animal. Hence, the non-random segregation of mitochondria represents an intrinsic mechanism for cell fate divergence in the context of asymmetric cell division in vivo.

## Methods
### Strains and genetics
Unless noted otherwise, all *C. elegans* strains were cultured at 20 °C as described[44]. Bristol N2 was used as the wild-type strain. Mutations and transgenes used in this study are: LGII: *fzo-1(tm1133)*[45]. LGIII: *rdvIs1*[28]. LGIV*: ced-3(n2427)*[46], *ced-4(n717)*[47], *drp-1(bc455)* (this study), *drp-1(tm1108)*[45], *drp-1(dx230)* (DRP-1(*dx230*Internal))[25], *pig-1(gm344)*[26], *bcIs153*[23] (P*toe-2*::mtGFP::unc-54 3'UTR[23] + P*egl-17*Myri-mCherry::pie-1 3'UTR[28] + P*egl-17*mCherry-TEV-S::his-24[28] + rol-6(su1006)[48]), *bcIs158* (this

study) [P$_{toe-2}$::mtGFP::unc-54 3'UTR[23] + P$_{egl-17}$Myri-SFmTurquoise2ox::pie-1 3'UTR[23] + P$_{egl-17}$SFmTurquoise2ox-TEV-S::his-24[23] + rol-6(su1006)[48]], and bcIs133[32]. Additional transgenes used in this study are: bcIs159 (this study) [P$_{mab-5}$EYFP-CUP-5::unc-54 3'UTR (this study) + P$_{egl-17}$Myri-SFmTurquoise2ox::pie-1 3'UTR + P$_{egl-17}$SFmTurquoise2ox-TEV-S::his-24 + rol-6(su1006)[48]], bcIs160 (this study) [P$_{toe-2}$::mtGFP::unc-54 3'UTR + P$_{toe-2}$SP-12mCherryKDEL::unc-54 3'UTR (this study) + P$_{egl-17}$Myri-SFmTurquoise2ox::pie-1 3'UTR + P$_{egl-17}$SFmTurquoise2ox-TEV-S::his-24 + rol-6(su1006)[48]] and bcIs162 (this study) [P$_{toe-2}$TOMM-20(N-term)-mKate2::tbb-2 3'UTR[23] + P$_{egl-17}$Myri-SFmTurquoise2ox::pie-1 3'UTR + P$_{egl-17}$SFmTurquoise2ox -TEV-S::his-24 + rol-6(su1006)[48]], bcEx1422 (P$_{eft-3}$2xCOX8A(N-term1-25aa)::HyPer7::unc-54 3'UTR[29]) (this study). Throughout our studies, we used information and tools available on WormBase (https://wormbase.org/#012-34-5)[49,50].

## Plasmid construction

All plasmids, inserts and oligos used are listed below (end of Methods section). Amplicons were generated through Q5® High-Fidelity 2X Master Mix (NEB, cat. # M0492S) PCR (CDS) or LongAmp® Taq 2X Master Mix (NEB, cat. # M0287S) PCR (promoter sequences). Gibson assembly was performed with the NEBuilder® HiFi DNA Assembly Master Mix (NEB cat. # E2621). **pBC1978** (P$_{mab-5}$EYFP::cup-5::unc-54 3'UTR): the plasmid pBC1977 (P$_{mab-5}$unc-54-3'UTR) was digested with NheI and BstEII to generate the vector backbone. The EYFP insert was amplified from pBC1689 template with oligos IS-EYFP-N-term-F and IS-EYFP-N-term-R. The cup-5 insert was amplified from pBC1983 (P$_{mab-5}$cup-5::EYFP::unc-54 3'UTR) with oligos CUP-5-for-Nterm-EYFP-F and CUP-5-for-Nterm-EYFP-R. The two inserts were inserted into the vector backbone through Gibson Assembly. The cup-5 CDS was inserted into pBC1983 by removing its 5$^{th}$ intron (2659 bp), between exons 5 and 6, containing the uncharacterized gene R13A5.15. **pBC1984** (P$_{toe-2}$SP12-mCherry-KDEL::unc-54 3'UTR): the plasmid pBC1590 was digested with NheI and NotI to generate the vector backbone. The SP12-mCherry-KDEL insert was amplified from pBC986 template with oligos NheI-For and NotI-Rev and then digested with NheI and NotI. The insert was inserted into the vector backbone through T4 ligation (T4 DNA ligase (NEB, cat. # M0202).

## CRISPR-Cas9-mediated generation of the bc455 allele

In order to CRISPR-Cas9 the drp-1(tm1108) deletion, we adapted a recently published methodology[51]. All RNA molecules are listed below (end of Methods section). The tm1108 deletion variant was generated by selecting two crRNAs named ISsgRNA1(tm1108) and ISsgR-NA2(tm1108) that were identified using CRISPOR 5.01 (input sequence ce11-chrIV-5538489-5540480, 1991bp long, PAM: NGG). The repair template IStm1108REPAIRDNA was designed such that both "arms" of the ssDNA had 40–50 bp overlap (lower case) with both blunt ends after cutting out the deletion sequence, while the central part (upper case) consists of a small duplication of the 5' end that naturally occurred during the original TMP/UV mutagenesis of drp-1[45]. The protocol consisted of three main steps prior to microinjection. Preparation of MIX1 (incubation at 95 °C for 1 minute + cooling): 50 µM 5' crRNA (ISsgRNA1(tm1108)) (2 µl), 50 µM 5' crRNA (ISgsRNA2(tm1108)) (2 µl) and 100 µM tracrRNA (2 µl). Preparation of MIX2 (incubation at 37 °C for 15 minute): MIX1 (1.5 µl), Cas9 (62 µM) (0.4 µl) and dpy-10 crRNA (50 µM) (1 µl). Preparation of MIX3, which was spun down at 12,000 × g for 3 minutes and the top 7 µl were transferred into a new tube as the final microinjection solution: MIX2 (2.9 µl), IStm1108RE-PAIRDNA (1 µl) and 1× TE, pH 7.5 (IDT) (5.5 µl). The MIX2 contains the dpy-10 crRNA as a coinjection marker to select those animals (dumpy phenotype) in which CRISPR-Cas9 gene editing was successful[52]. Animals were outcrossed once with N2 (wild-type) animals to remove the dpy-10 mutation in homozygosity (bcIs158) or self-crossed to remove the dpy-10 mutation in heterozygosity (bcIs153). crRNAs were synthesized by Merck with the addition of a 5' modification (2'-O-methyl and

phosphorothioate linkages) for extra stability and purified by HPLC. crRNAs were ordered without premixing with tracrRNA. Cas9 and tracrRNA were synthesized by IDT.

## Mechanical immobilization of L1 larvae

The immobilization and imaging protocols used to study QL.p division in wild type, pig-1(gm344), fzo-1(tm1133) and drp-1(bc455) in super-resolution and at high temporal resolution (Fig. 2 and related figures) are the same as previously reported[23]. For bcIs160 and bcIs159; bcIs162 animals (Fig. S1), the immobilization protocol is the same as previously reported[23].

## Image processing and 3D rendering from recordings acquired on animals mechanically immobilized

The image processing protocol is the same as previously reported[23], but performing 5 iterations of the Richardson-Lucy deconvolution algorithm instead of 20 iterations. We observed that reducing the number of iterations reduced the risk of artificial fission of mitochondrial volumes during 3D rendering in Imaris. DRP-1 foci were rendered following the same pipeline with the following changes: using the theoretical PSF of mNeongreen rather than GFP, and the Imaris spots algorithm was used rather than its surface algorithm. For the spots algorithm, the xy diameter was set at 0.353 µm, while the estimated PSF-elongation along the z-axis was set to double that (0.706 µm), segmenting into ellipsoid spots. The quality threshold for spots for all samples was set at 3.

## Confocal microscopy and 2D image analysis (mitochondria + ER experiment)

We used an upright LSM980 with an Airyscan2 setup with a GaAsP-TMP detector and performed Super Resolution Airyscan with a C Plan Apochromat ×63/1.40 objective. mCherry, GFP and SFmTorquoise2ox fluorophores were excited respectively through 594 nm (0.5%), 488 nm (0.2%), and 405 nm (0.5%) wavelengths, while their detection ranges were always 300–720 nm. We adjusted other settings as follows: FOV = 13.47 × 13.47 µm (382 × 382 pixels), voxel size = 0.035 × 0.035 × 0.130 µm, pixel time = 0.66 µs (frame time = 449.4 ms), detector gain = 850 V, and scan direction = bidirectional (no averaging). Z-stacks were Airyscan-processed (2D, standard strength) before export. We did not perform a pairwise acquisition of metaphase and post-cytokinesis for the ER images, due to suspected phototoxicity of SP12-mCherry-KDEL and mtGFP, resulting in the appearance of circular mitochondria during post-cytokinesis (images not shown). Images were then processed in Fiji (ImageJ v1.53f51) through background subtraction (rolling ball radius = 60 pixels). We drew ROIs along the cell boundary on each slice of the z-stack, and we measured area and Integrated Density within each ROI and summed them up (for all slices relative to the QL.p anterior side, QL.p posterior side, QL.pa and QL.pp). We then calculated the mean Integrated Density (integrated density/area) from summed areas and Integrated Densities to obtain the overall density of fluorescence intensities (a proxy for organelle density) at both metaphase and post-cytokinesis. The summed areas were used to calculate the respective cell size at both metaphase and post-cytokinesis. We used summed areas and densities of fluorescence intensities to measure organelle ratios and cell volume ratios at metaphase and post-cytokinesis.

## Validation of the lysosome-specific marker

We validated the lysosome-specificity of the EYFP::CUP-5 marker by co-labeling lysosomes with lysotracker Red DND-99® (Thermo Fisher Scientific, cat. # L7528) (Fig. S2). Not all lysotracker-labeled lysosomes are visible with EYFP::CUP-5 because transcription of this reporter gene is driven by the promoter region of mab-5. This gene encodes a homeobox protein (TF) that controls the formation of the posterior-specific pattern of cells during the post-embryonic development of C.

*elegans*[53]. *mab-5* is expressed only in a subset of cells found in the posterior half of L1 larvae, among which QL descendants[54,55]. We calculated the Manders colocalization coefficient (MCC) of EYFP::CUP-5 fluorescence with that of lysotracker Red. On average, the unthresholded MCC was 0.91 (sd ± 0.02), while the thresholded MCC was 0.81 (sd±0.16), which is similar to the MCC previously obtained for cemOrange::cup-5[56].

## Confocal microscopy and 2D image analysis (mitochondria + lysosome experiment)

We used an upright LSM980 with an Airyscan2 setup and a GaAsP-TMP detector and performed Super Resolution Airyscan with a C Plan Apochromat ×63/1.40 objective. mKate2, EYFP and SFmTorquoise2ox fluorophores were excited respectively through 594 nm (0.4%), 514 nm (0.7%), and 445 nm (0.7%) wavelengths, while their detection ranges were always 300–720 nm. We adjusted other settings as follows: FOV = 12.54 × 12.54 μm (324 × 324 pixels), voxel size = 0.039 × 0.039 × 0.130 μm, pixel time = 0.72 μs (frame time = 409.59 ms), detector gain = 850 V, and scan direction = bidirectional (no averaging). Z-stacks were Airyscan-processed (2D, standard strength) before export. We performed a pairwise acquisition of metaphase and post-cytokinesis images, which were then processed in Fiji through background subtraction (rolling ball radius = 60 pixels). We drew ROIs along the cell boundary on each slice of the z-stack, and we measured area and Integrated Density within each ROI and summed them up (for all slices relative to the QL.p anterior side, QL.p posterior side, QL.pa and QL.pp). We then calculated the mean Integrated Density (integrated density/area) from summed areas and Integrated Densities to obtain the overall density of fluorescence intensities (as a proxy of organelle density) at both metaphase and post-cytokinesis. The summed areas were used to calculate the respective cell size at both metaphase and post-cytokinesis. We used summed areas and density of fluorescence intensities to measure organelle ratios and cell volume ratios at metaphase and post-cytokinesis.

## TMRE staining of L1 larvae

NGM plates with TMRE were prepared one day prior to imaging. NGM agar was melted, and TMRE (Invitrogen™, cat. # T669) was added to a final concentration of 10 nM. NGM + TMRE was then poured into small petri plates (±1/2 ml) and left to cool for 2–3 hours in the dark. Next, *E. coli* OP50 was suspended in M9 and pipetted on plates (20 µl). L1s animals expressing *bcIs158* were synchronized for 1 hr by washing out all adults and letting eggs hatch for 1 hr on medium NGM plates. Freshly hatched L1s were washed off the plates with MPEG and pipetted on bacterial lawns grown overnight on NGM + TMRE plates. After 2.5 hours of growth at 20 °C, L1s were washed off and transferred to a seeded medium NGM plate without TMRE for another 2.5 hours (or longer for slow-growing strains) at 20 °C. Mounting and immobilization were performed following the protocol outlined in our previous study[23].

## Confocal microscopy and 3D image analysis (TMRE experiment)

We used an upright LSM980 with an Airyscan2 setup and a GaAsP-TMP detector and performed Super Resolution Airyscan with a C Plan Apochromat ×63/1.40 objective. TMRE, GFP, and SFmTorquoise2ox fluorophores were excited at 561 nm (1.0%), 488 nm (0.9%), and 405 nm (0.8%) respectively, while their detection ranges were always 300–720 nm. We adjusted other settings as follows: FOV = 12.62 × 12.62 μm (358 × 358 pixels), voxel size = 0.035 × 0.035 × 0.130 μm, pixel time = 0.66 μs (frame time = 449.4 ms), detector gain = 850 V, and scan direction = bidirectional (no averaging). Z-stacks were Airyscan-processed (2D, standard strength) before export. We did not perform a pairwise acquisition of metaphase and post-cytokinesis images due to suspected phototoxicity of TMRE and mtGFP, resulting in the occasional appearance of circular mitochondria post-cytokinesis (images

not shown). Images of the mtGFP channel in Fiji only were processed through two consecutive background subtractions (rolling ball radius = 50 pixels). Average TMRE fluorescence values were calculated per cell from actual TMRE fluorescence within mitochondrial volumes in Imaris and were used to compare different genetic backgrounds. Since L1 larvae absorbed the TMRE dye at different levels due to inter-animal variability in feeding and metabolic activity, mitochondria of QL.p in distinct animals are not comparable across samples, without normalization. Therefore, comparisons between mitochondria within individual cells were performed by normalizing TMRE fluorescence (=TMRE fluorescence that is relative to the respective TMRE average in QL.p at metaphase or at post-cytokinesis). The normalization was done by measuring the average TMRE fluorescence per cell (overall average of QL.p anterior and QL.p posterior sides (metaphase) or overall average of QL.pa and QL.pp combined (post-cytokinesis)), and the normalized TMRE fluorescence was calculated as: (individual TMRE fluorescence/average TMRE fluorescence)*100. Next, the normalized TMRE fluorescence was paired with the respective mitochondrion volume and sphericity. Finally, we classified mitochondria by splitting the min-max volume range of the control data sets (metaphase and post-cytokinesis) into 6 morphological classes (class 1: round (=fragmented) mitochondria; class 6: tubular/networked (=fused) mitochondria. The same volume intervals were applied to mutant data sets.

## Confocal microscopy and 3D image analysis (HyPer7 experiment)

We used an upright LSM980 with an Airyscan2 setup with a GaAsP-TMP detector and performed Super Resolution Airyscan with a C Plan Apochromat ×63/1.40 objective. HyPer7 (violet excitation), HyPer7 (green excitation), and mCherry fluorophores were excited at 405 nm (0.6%), 488 nm (0.6%), and 594 nm (0.6%), respectively, and their detection ranges were 300–720 nm. We adjusted other settings as follows: FOV = 12.62 × 12.62 μm (358 × 358 pixels), voxel size = 0.035 × 0.035 × 0.130 μm, pixel time = 0.66 μs (frame time = 449.4 ms), detector gain = 850 V, and scan direction = bidirectional (no averaging). Z-stacks were Airyscan-processed (2D, standard strength) before export. We performed pairwise acquisitions of metaphase and post-cytokinesis images, as we did not observe abnormal mitochondrial morphologies at either time point. Images were aligned (MultiStackReg Plugin–Fiji) according to the reference transformation file generated on the mCherry channel. Next, the HyPer7 (green excitation) was corrected by subtracting the mCherry image fluorescence according to mCherry excitability under 488 nm irradiation, as previously done[21]. Next, the HyPer7 (violet excitation) and HyPer7 (green excitation) channels were summed up to generate a fourth image channel. This fourth channel was processed through two consecutive background subtractions (rolling ball radius = 50 pixels) and then used in Imaris to 3D render mitochondria without biasing for any of the HyPer7 channels. Next, mCherry images were used to draw ROIs along the cell membrane throughout the z-stack portion intersecting cells, both at metaphase and post-cytokinesis. Those ROIs were used to remove fluorescence intensities outside the ROIs as described[21]. Using this approach, mitochondria of neighboring cells do not interfere with the segmentation of mitochondria in QL.p. Four-channel images were then opened in Imaris, and mitochondria were automatically 3D rendered using the fourth channel (HyPer7 (violet excitation)+HyPer7 (green excitation)) and mean HyPer7 (violet excitation) and HyPer7 (green excitation) fluorescence intensities were extracted from each mitochondrion as well as mitochondrial volume and sphericity. Finally, we classified mitochondria by splitting the min-max volume range of the control data sets (metaphase and post-cytokinesis) into 6 morphological classes as described for the TMRE analysis (see above). We generated HyPer7 488/405 ratiometric values as described[29].

## Microfluidics: L1 larvae preparation

For all experiments, we adapted the methodology described in *Berger* et al. *(2021)*[22] for use with L1 larvae.

**L1 larvae preparation.** L1 larvae were prepared by washing out all adults and larvae from medium plates (4× or more plates) with MPEG (5.8 g Na2HPO4, 3 g KH2PO4, 0.5 g NaCl, 1 g NH4Cl to 1 l MilliQ water and autoclave, then add 1 g Polyethylene glycol 8000 (MPEG 0.1%)). We let embryos hatch for 90 minutes and collected L1s by washing plates with MPEG (2 ml for 4–5 plates by transferring buffer+L1s plate-by-plate) into a 1.5 ml microcentrifuge tube(s). We next filtered L1s through a 10 μm (pluriStrainer Mini 10 μm, PluriSelect) and pipetted the filtered MPEG+L1s into 1.5 ml tubes. We then centrifuged animals at $400 \times g$ for 3 min and discarded the MPEG leaving L1s in the pellet (-20 μl). We next added 1.7 ml of S-Basal and centrifuged at 400 x g for 3 min. Subsequently, we discarded the S-Basal, leaving 50/60 μl with pelleted L1s, which were used for microfluidics mounting.

**Food preparation.** We added 0.2 μm-filtered LB in a 50 ml Falcon tube to 20 ml final volume (2× falcons = 40 ml). Next, we added 45 μl of *E. coli* NA22 culture ($OD_{600}$ = 1.9; kept at 4 °C) to both Falcon tubes and let them grow for 16.5 hrs in a 37 °C shaking incubator (final $OD_{600}$ was always between 1.7–2.0). The $OD_{600}$ was measured using a spectrophotometer (Thermo Scientific Genesis 30 Visible). Bacteria were spun down at $3000 \times g$ (or rcf) (15.3 rad) for 5 min (Eppendorf Centrifuge 5804 R). We discarded the liquid and resuspended the pellet in 2 ml of S-Basal by pipetting up and down, and then combined the two 2 ml volumes. We spun down the tube at $3000 \times g$ for 3 min, discarded the liquid and resuspended the pellet in 4 ml of S-Basal. We repeated this step once more. We then resuspended the pellet in 1 ml S-Basal and transferred this into a 2 ml microcentrifuge tube. We added OptiPrep (Sigma-Aldrich, cat. # D1556) (0.65 ml) and S-Basal+Pluronic F127 (1% w/v) (Sigma-Aldrich, cat. # P2443) (0.33 ml) to the bacterial suspension and mixed by gently pipetting up and down. (Fast pipetting can make the bacterial preparation toxic for L1 animals.) We next filtered the *E. coli* NA22 solution twice, once through a 10 μm filter and next through a 5 μm filter (pluriStrainer Mini 10 μm or 5 μm, PluriSelect, HS code 39269097 and 39269097).

**Microfluidics device operation.** We mounted L1 larvae in the L1 chip as described before[22] and followed their development throughout the experiment (up to 22 hours). We manually switched between Multiplex Airyscan imaging (see below) and Airyscan Super Resolution (see next two sections) whenever QL.p and QL.pa were at metaphase and at post-cytokinesis. To achieve super-resolution image quality, we manually actuated the on-chip compression layer (20 psi and 15 psi during QL.p and QL.pa divisions, respectively) prior to switching to super-resolution mode. We did not compress the system at any other time (i.e., while following the cell in Multiplex Airyscan mode) to minimize mechanical stress.

## Characterization of QL.pp cell death

Using the adapted methodology originally described in Berger et al.[22], we imaged wild-type animals expressing the *bcIs153* transgene (Fig. S14a, b). No compression was applied during the entire recording, and images were taken every 5.5 minutes for 12 hours total. We used an inverted LSM980 with an Airyscan2 setup with a GaAsP-TMP detector (LSM) and Multialkali-PMT detector (DIC). We performed LSM + DIC (Nomarsky) with a C Plan Apochromat ×63/1.40 objective. We set the High Contrast (DIC III) Wollastone prism for DIC and manually set all microscope components (aperture, condenser, polarizer and Wollastone prism) to optimize DIC image quality. LSM mode was selected for acquisition of the mCherry signal (561 nm excitation wavelength, laser power = 0.08%), while DIC images were acquired through a transmission channel (T-PMT) using the same wavelength used to excite mCherry. The detection wavelengths for mCherry and the transmitted channels are 570–694 nm and 300–900 nm, respectively. We adjusted other settings as follows: FOV = 133.46 × 34.57 μm (2965 × 768 pixels), voxel size = 0.045 × 0.045 × 1.000 μm, pixel time = 0.26 μs (frame time = 449.74 ms), detector gain = 610 V, and scan direction = bidirectional (no averaging). We used DIC and LSM to follow morphological changes of QL.pp, including its transformation into a refractile cell corpse[17]. We identified a sudden morphological shift from shrunk round QL.pp to prolate spheroid corpse, which coincided with the appearance of refractility by DIC (Fig. S15c, white arrowheads). This morphological change was followed by the quick disappearance of the corpse (in LSM) (Fig. S15c, times +187 min to +209 min). We investigated multiplex confocal recordings and identified the exact same change (Fig. S15d, times +147 min and +150:30 min). QL.pp fluorescence follows a unimodal distribution before death (Fig. S15e, black profiles) and a multimodal distribution after death (Fig. S15e, red profiles). These fluorescence distributions coincide with the respective morphologies (Fig. S15f). This change in QL.pp morphology was used to determine the time of QL.pp death.

## Confocal microscopy to follow QL.pa and QL.pp fates

For this experiment, we followed the methodology described in Berger et al.[22] with the changes mentioned above. We used wild-type and mutant animals expressing the *bcIs153* transgene. We used an inverted LSM980 with Airyscan2 to image wild-type and *drp-1(bc455)* animals and an upright LSM980 with Airyscan2 to image *fzo-1(tm1133)*, due to the unavailability of the inverted microscope during the experiment. We followed the entire QL.p lineage through Multiplex Airyscan SR-8Y (long-term imaging) imaging and acquired Airyscan super resolution images (see below) for metaphase and post-cytokinesis during both QL.p and QL.pa cell divisions. We performed Multiplex Airyscan with a C Plan Apochromat ×63/1.40 objective using the Q lineage markers (mCherry) only. mCherry was excited at 561 nm (laser power = 0.3% (=0.03% in standard LSM mode)), and the detection wavelength was 574–720 nm. The tile function was used to save multiple XYZ positions so that positions (P1, P2, P3, …) were organized and assigned across the microfluidic device geometry. We adjusted other settings as follows: FOV = 133.46 × 34.57 μm (1992 × 516 pixels), voxel size = 0.067 × 0.067 × 1.000 μm, pixel time = 1.05 μs (frame time = 159.74 ms), detector gain = 990 V, and scan direction = bidirectional (no averaging). After completing the experiment, recordings were 2D Airyscan-process (standard strength) in ZenBlue 3.3. Airyscan-processing was necessary to visualize and process images in ImageJ.

## Confocal microscopy to record QL.p and QL.pa divisions in super-resolution

We used an inverted or upright LSM980 with an Airyscan2 (see above) setup with a GaAsP-TMP detector and performed Super Resolution Airyscan with a C Plan Apochromat ×63/1.40 objective. mCherry and GFP fluorophores were excited at 561 nm (0.6%) and 488 nm (0.6%), respectively, while their detection ranges were always 300–720 nm. We adjusted other settings as it follows: FOV = 20.66 × 7.12 μm (464 × 160 pixels), voxel size = 0.045 × 0.045 × 0.130 μm, pixel time = 1.08 μs (frame time = 226.89 ms), detector gain = 990 V, and scan direction = bidirectional (no averaging). Z-stacks were Airyscan-processed (2D, standard strength) before export. Images were processed in Fiji and 3D rendered in Imaris according to our previous study[23], with the following changes. Before channel subtraction (mtGFP channel – mCherry channel), the mCherry image was multiplied by 0.08 to reproduce the theoretical emission of mCherry (8%) when excited by the 488 nm wavelength. Images were deconvolved using 10 iterations of the Richardson-Lucy algorithm.

## QL.pp survival assay

The number of surviving QL.pp cells was scored using the $P_{toe-2}gfp$ (*bcls133*) transgene, as previously described[32]. Late L1 larvae/young L2 larvae (>10 hours post-QL.p division) were mounted on 2% agar pads using 10 mM levamisole in M9 buffer as a paralytic agent. The number of GFP-positive cells was determined using a Zeiss Axioscope 2. In wild-type worms, there are two GFP-positive cells representing the QL.pa daughters (PVM and SDQL neurons). Up to two extra GFP-positive cells can be seen in mutants, representing inappropriately surviving QL.pp or QL.pp daughters. To validate the counting, all GFP-positive cells were assessed in DIC to ensure they were not GFP-positive corpses. The QL.pp survival percentage represents the number of QL.pp that inappropriately survived (animals with 1 or 2 extra GFP-positive cells) divided by the sample size (number of animals analyzed).

## Statistical analysis

Statistical analyses were performed using GraphPad Prism8 and R (v4.3.2). For two-sample comparisons of organelle densities and organelle density ratios, student's *t*-tests and Wilcoxon's tests were used based on Shapiro-Wilk's normality test to check the normality assumption. Where ties existed for Wilcoxon's signed-rank, the Pratt method (coin package)[57]. Pairwise t-tests and signed-rank test were used for paired data and are indicated in the figures by lines connecting paired data instead of points. Spearman's rank correlation coefficient was used to test for the strength of monotonic relationships throughout.

For the principal component analysis, six variables of mitochondrial morphology (volume, surface area, sphericity, surface to volume ratio, and oblate and prolate ellipticity) were averaged for each cell and side of the cell (i.e., QL.p anterior, QL.p posterior, QL.pa, and QL.pp). These were then scaled to have unit variance, and principal components were then constructed with PC1 containing mostly size variation, PC2 containing predominantly variance in ellipticity, and PC3 containing predominantly variance in sphericity/surface to volume ratio. The first three principal components always contained at least 92% of all variation for all genotypes studied; the remaining principal components were not considered for analysis. To test for differences following this exploratory analysis, linear models were constructed with anteroposterior location (or QL.pa-QL.pp) as a factor, residuals were inspected visually, and analysis of variance was carried out.

For the QL.pp survival assay (cell counts), one-sided Fisher's exact tests were used to compare relevant genotypes and the false discovery rate was used to correct for multiple testing. For the survival analysis of the microfluidics imaging, the time until death was recorded. As the five surviving cells could not be followed indefinitely, the data were considered right-censored as a consequence. Cox proportional-Hazard's models were fitted using the *survival* package[58,59], and goodness of fit was explored graphically using the *survminer* package[60]. The Schoenfeld and Martingale residuals were explored visually, as well as explicitly tested for. The eventual Cox regression model correlates survival time in minutes, by genotype and with mitochondrial density as a covariate using smoothing splines with three degrees of freedom. The latter was done to satisfy the linearity assumption of covariates in the Cox proportional-Hazards model. This term, the mitochondrial density spline, is visualized as the relative death risk by mitochondrial density in Fig. 5a.

## Figures and illustrations

Figures and schematics were generated in Affinity Designer 1.7.3, GraphPad Prism8 and R 4.3.2, and the Kaplan–Meier curves were plotted using the ggplot2 package[61].

## List of oligos and RNA molecules used for cloning and mutagenesis in this study

IS-EYFP-N-term-F: ttaataactaaaatattatcgctagctttcagatggtgagcaagggcgagga

IS-EYFP-N-term-R: atcagttgtgctccggcgagacatgcctccggcgccagcacctgcgccctgtacagctcgtccatgccc

CUP-5-for-Nterm-EYFP-F: atgtctcgccggagcacaac

CUP-5-for-Nterm-EYFP-R: caattctacgaatgctattggttaccctactcgagtcgttgccagccg

NheI-For: taatctggctagcaaaaatgcataagg

NotI-Rev: gcggtggcggccgctcacagctcatccttatacaat

ISsgRNA1(tm1108): GCUCCGAAGUAGCGAAAUCC

ISsgRNA2(tm1108): CGAAUGCAACAAGAUCCGAU

Stm1108REPAIRDNA: ttcaaatccttcttctatcattctggctgtaactccagcgaaccaggaCTCCAGCGAACCAGGATTtcggatcttgttgcattcggtgaaccagttgaagataagaac

dpy-10 sgRNA: GCUACCAUAGGCACCACGAG

## Reporting summary

Further information on research design is available in the Nature Portfolio Reporting Summary linked to this article.

## Data availability

All data and materials used in this manuscript are available and can be requested from the corresponding author. Source data are provided with this paper.

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

## Acknowledgements

We thank members of the Conradt lab, the Center for Cell and Molecular Dynamics (https://www.uclccmd.co.uk/) and T. Schedl for discussions and comments on the manuscript. We thank L. McGuinness for excellent technical support. Some strains were provided by the *Caenorhabditis* Genetics Center (CGC), which is funded by NIH Office of Research Infrastructure Programs (P40 OD010440). We thank Alex Hajnal (University of Zurich, Switzerland) and Andrew deMello (ETH Zurich, Switzerland) for their support of S.B. This work was supported by a predoctoral fellowship from the Studienstiftung des deutschen Volkes to NM, funds from UCL (Division of Biosciences, UCL LSM Capital Equipment Fund) to B.C., and a Wolfson Fellowship from the Royal Society (https://royalsociety.org/) to B.C. (RSWF\R1\180008), and the Biotechnology and Biological Sciences Research Council (https://bbsrc.ukri.org/) (BB/V007572/1 and BB/V015648/1to B.C.).

## Author contributions

Experiments were performed by I.S. and J.V.E. S.B., N.M., and E.J.L. provided resources and methodology. I.S., J.V.E., E.J.L., and B.C. participated in designing experiments, data analysis, and data interpretation. I.S. and B.C. wrote the manuscript. All authors (I.S., J.V.E., S.B., N.M., E.J.L., and B.C.) provided input and revisions to successive drafts of the entire manuscript. B.C. managed the overall project and obtained funding.

## Competing interests

The authors declare no competing interests.
