## [Peer Review file · Nature Communications]

Unequal segregation of mitochondria during asymmetric cell division contributes to cell fate divergence in sister cells in vivo

Corresponding Author: Professor Barbara Conradt

Version 0:

Reviewer comments:

Reviewer #1

(Remarks to the Author)

Segos et al. presented an elegant study describing the high-resolution distribution of mitochondria in live *C. elegans* during asymmetric cell division. They compared the outcomes of mitochondrial distribution with those of other organelles and discovered that mitochondrial distribution is uneven. The cell that survives and continues to divide exhibits greater mitochondrial density than the cell that dies post-division. Other organelles examined, such as lysosomes and the ER, exhibited more subtle distribution differences. For these studies, the authors included solid controls, including multiple mitochondrial markers. They also found that modulating mitochondrial dynamics (fusion/fission) impacts mitochondrial segregation and cell death fate. This finding is interesting and significant to the field. While the data presented show a clear correlation between mitochondrial density and cell outcomes, the mechanism remains somewhat underexplored. Additional avenues could be explored to strengthen the conclusion that mitochondrial differences drive distinct cellular outcomes. For example, are the mitochondria inherited by the cell that dies also better primed to induce apoptosis? Given that mitochondrial potential is distinct between QL.pa and QL.pp, it remains unclear whether directly modulating mitochondrial OXPHOS influences cell death outcomes. Moreover, other biochemical properties, such as ROS, which could dictate the death fate, remain unexplored.

Reviewer #2

(Remarks to the Author)

In their study, Segos et al. utilize live imaging and super-resolution microscopy to report mitochondrial segregation during asymmetric cell division in *C. elegans* larvae. Specifically, during the asymmetric division of neuroblast QL.p, which generates a smaller daughter cell (QL.pp) and a larger daughter cell (QL.pa), they show that mitochondria segregate unequally in terms of density and morphology, a process dependent on mitochondrial fission and fusion dynamics. The study also finds a positive correlation between mitochondrial density in QL.pp and the time it takes for QL.pp to undergo cell death. While the use of super-resolution microscopy is used impressively to demonstrate asymmetric mitochondrial segregation and the study adds importantly to the growing body of evidence on organelle segregation patterns during asymmetric cell divisions, the study does not provide new direct and mechanistic evidence on the segregation mechanism or cell fate regulation in neuroblasts.

Specific Comments

1. Mitochondrial fusion and fission events were visually quantified and authors claim that fission occurs significantly more often on the posterior side. Did the authors observe increased Drp1 localization on mitochondria in the posterior region? To strengthen the claim authors should Quantify Drp1 localization on mitochondria in the posterior vs. anterior sides.
2. Authors state: "our findings are consistent with the notion that unequal mitochondrial segregation during QL.p division and, hence mitochondrial density in QL.pp, correlates with the acquisition of the cell death fate by QL.pp." As the evidence is correlative, the authors are to be applauded for arguing on this notion very carefully. However, the manuscript would greatly benefit from additional mechanistic insights. The key would be to address whether the asymmetric apportioning of mitochondria actually predisposes the cells to die. Do the mitochondria in QL.pp cells exhibit for example markers of damage?

3. It is unclear whether disrupting mitochondrial asymmetry in the mutant conditions affects the fate acquisition of the daughter cells. Is the acquisition of the cell death fate by QL.pp also disrupted?
4. Mitochondria are classified into distinct classes based on their shape; however, the authors do not provide any data demonstrating that these mitochondria are functionally distinct. For example, they could examine the ultrastructure of these mitochondria, especially since they do not observe any differences in mitochondrial membrane potential (MMP).

Minor comments:

"Post-cytokinesis, the two daughter cells generated were unequal in size 1 with QL.pa/QL.pp size ratios of 3.33 (lysosomes) and 3.25 (ER) (Fig. S1c, d, right)." The numbers presented in the text do not completely align with those shown in the figure.

The introduction provides an inaccurate portrayal of previous works and some key studies in the field are omitted. As a result, the authors claim for example that "Importantly, none of the studies in mammalian cells analysed mitochondria quantitatively and in real time as cells divide." This is not true, and authors should also edit the text to reduce redundant statements. Moreover, some of the earlier work actually address mitochondria and other sub cellular compartments similarly to this study, and results should be discussed accordingly.

Reviewer #3

(Remarks to the Author)

It has been shown that asymmetric segregation of mitochondria during cell division determines the fate of daughter cells in aging yeast cells and mammalian stem-like cell lines. It is very likely that this is a more general mechanism, which also is important in metazoan development. Here, Segos et al. present the first in vivo evidence for this in living animals. Using *C. elegans* larvae they show that mitochondria segregate unequally during asymmetric division of a neuroblast that produces one daughter destined to differentiate and another daughter destined to die. The cell destined to die is much smaller and receives fewer mitochondria (also in relation to the cell volume), which are more fragmented. Interference with fusion/fission dynamics abrogates asymmetric inheritance of mitochondria. This correlates with the generation of some cells that inappropriately survive. Hence, the authors propose that non-random segregation of mitochondria contributes to cell fate divergence during asymmetric cell division. This is a beautiful manuscript. The hypotheses are well defined, the data are very clear and well controlled, the microscopy is impressive, and the manuscript is well written. I have only a few points that should be considered by the authors.

Important points:

1. Figure 2 shows that mitochondria are more fragmented in QL.pp than in QL.pa. It would be important to know whether this morphological difference emerges already before or only after cytokinesis. In other words: What is the evidence that mitochondrial fragmentation in the posterior half of QL.p starts already before the cytosols of the two daughter cells are separated? Figure S5 quantifies fusion/fission events from metaphase to post-cytokinesis. Can these data be shown separately for pre and post-cytokinetic cells?
2. The Conradt lab has reported previously that block of DRP-1 activity causes inappropriate survival of cells during embryonic development by blocking apoptosis. How can the authors exclude that inappropriate survival of QL.pp cells in *drp-1* (lf) animals is exclusively due to interference with the apoptotic machinery rather than abnormal segregation of mitochondria? This should be discussed more extensively.

Minor points:

3. Figure S1: Why did the authors choose different ways of displaying their data in Figure S1 (lines versus dots)?
4. As fission and fusion must be balanced to maintain tubular mitochondria, it would make sense to indicate the fusion/fission ratios in Figure S5 in addition to the absolute numbers.
5. Cell volume ratios should be added for the *pig-1* mutant, which is supposed to divide symmetrically.
6. The order of the panels in the figures should be the same as the order they are mentioned in the text (for example Fig. 3b is mentioned in the text only after Fig. 3e).
7. Figure 3e: *fzo-1* mutants should have fragmented mitochondria and *drp-1* mutants should have hyperfused mitochondria. Why are some class 6 mitochondria found in *fzo-1* and class 1 mitochondria in *drp-1*?
8. The authors should briefly explain in the text what a multivariate Cox proportional hazards regression is (first mentioned in line 348).

Version 1:

Reviewer comments:

Reviewer #1

(Remarks to the Author)

Segos et al. described the high-resolution distribution of mitochondria in live *C. elegans* during asymmetric cell division. They found that the cell that survives and continues to divide exhibits greater mitochondrial density than the cell that dies post-division. The technology development to carefully track organelle dynamics in live organisms during cell division is impressive. However, there is an enormous amount of literature that describes correlative studies on mitochondrial morphology and cellular outcomes. In this revision the authors do not advance our mechanistic understanding on mitochondrial density and apoptosis. The advantage of using *C. elegans* is the relatively quick development of tools to study mechanism. As such, I do not agree with the revision stating that question of mechanism is beyond the scope.

Reviewer #2

(Remarks to the Author)

I am in general pleased with the technical aspects of the revision work and the additional insights from the new experiments. However, the authors are consistently overlooking similar data (ROS, mitochondria membrane potential, DRP1) being probed and published in previous works, such as Katajisto et al. *Science*, 2015. Together with the authors for example ending their discussion by claiming: (line 587) "Hence, the non-random segregation of mitochondria represents a novel intrinsic mechanism for cell fate divergence in the context of asymmetric cell division", the manuscript suffers from overselling the conceptual advance it provides. Authors should clearly state how the fact that similar mechanisms have been probed previously and their data agrees to a large degree with previous works, but that here they focus on demonstrating the phenomenon in vivo.

Reviewer #3

(Remarks to the Author)

The authors have responded to my previous concerns in an adequate manner. Publication of this interesting manuscript can now be recommended.

REVIEWER COMMENTS

Reviewer #1 (Remarks to the Author):

Segos et al. presented an elegant study describing the high-resolution distribution of mitochondria in live *C. elegans* during asymmetric cell division. They compared the outcomes of mitochondrial distribution with those of other organelles and discovered that mitochondrial distribution is uneven. The cell that survives and continues to divide exhibits greater mitochondrial density than the cell that dies post-division. Other organelles examined, such as lysosomes and the ER, exhibited more subtle distribution differences. For these studies, the authors included solid controls, including multiple mitochondrial markers. They also found that modulating mitochondrial dynamics (fusion/fission) impacts mitochondrial segregation and cell death fate. This finding is interesting and significant to the field.

While the data presented show a clear correlation between mitochondrial density and cell outcomes, the mechanism remains somewhat underexplored. Additional avenues could be explored to strengthen the conclusion that mitochondrial differences drive distinct cellular outcomes. For example, are the mitochondria inherited by the cell that dies also better primed to induce apoptosis? Given that mitochondrial potential is distinct between QL.pa and QL.pp, it remains unclear whether directly modulating mitochondrial OXPHOS influences cell death outcomes. Moreover, other biochemical properties, such as ROS, which could dictate the death fate, remain unexplored.

Segos et al

We thank the reviewer for considering our finding interesting and significant to the field.

The reviewer states “the data presented show a clear correlation between mitochondrial density and cell outcomes, (but) the mechanism remains somewhat underexplored“. In addition, they ask “are the mitochondria inherited by the cell that dies also better primed to induce apoptosis?”. We agree with the reviewer that there are open questions. For example, we have not yet determined the mechanisms through which mitochondria impact apoptosis in QL.pp. One of our future goals is to uncover these mechanisms. To do so, we will need to conduct experiments at very high spatial and temporal resolution, which will require the development of new tools. For example, we will need to develop tools to visualise - in real time - components of the *C. elegans* apoptosis pathway at single molecule resolution and to knock-down gene function specifically in QL.pp. We consider the development of such tools beyond the scope of the current manuscript.

We also agree with the reviewer that it would be interesting to determine “whether directly modulating mitochondrial OXPHOS influences cell death outcomes“. Unfortunately, with currently available tools, we have so far been unable to conceive an experimental design that would allow us to combine the reduction of mitochondrial OXPHOS specifically in QL.pp with QL.pp fate tracking in the microfluidic system. Therefore, addressing this question will again require the development of new tools, which is beyond the scope of the current manuscript.

Finally, the reviewer states “Moreover, other biochemical properties, such as ROS, which could dictate the death fate, remain unexplored.” Using a genetically encoded sensor for mitochondrial H₂O₂, HyPer7, we have now determined mitochondrial ROS levels in QL.p and its daughter cells QL.pa and QL.pp. Our results demonstrate that there is no difference in ROS levels between mitochondria in QL.pa and QL.pp (see new paragraph “**Unequal mitochondrial segregation during QL.p division is not determined by mitochondrial ROS**“ in **Results (page 8)** and new **Figure S14**). Therefore, ROS levels are unlikely to play a role in determining unequal mitochondrial segregation and in priming mitochondria in QL.pp for apoptosis.

Reviewer #2 (Remarks to the Author):

In their study, Segos et al. utilize live imaging and super-resolution microscopy to report mitochondrial segregation during asymmetric cell division in *C. elegans* larvae. Specifically, during the asymmetric division of neuroblast QL.p, which generates a smaller daughter cell (QL.pp) and a larger daughter cell (QL.pa), they show that mitochondria segregate unequally in terms of density and morphology, a process dependent on mitochondrial fission and fusion dynamics. The study also finds a positive correlation between mitochondrial density in QL.pp and the time it takes for QL.pp to undergo cell death. While the use of super-resolution microscopy is used impressively to demonstrate asymmetric mitochondrial segregation and the study adds importantly to the growing body of evidence on organelle segregation patterns during asymmetric cell divisions, the study does not provide new direct and mechanistic evidence on the segregation mechanism or cell fate regulation in neuroblasts.

Segos et al

We thank the reviewer for considering our use of super-resolution microscopy impressive and for stating that our “study adds importantly to the growing body of evidence on organelle segregation”.

The reviewer also states: “the study does not provide new direct and mechanistic evidence on the segregation mechanism or cell fate regulation in neuroblasts.” To the best of our knowledge, our study is the first to report unequal mitochondrial segregation truly in an *in vivo* system and to present evidence that unequal mitochondrial segregation correlates with daughter cell fates. As stated above in our response to Reviewer 1, uncovering the mechanisms of unequal mitochondrial segregation in this *in vivo* system and determining the mechanisms through which mitochondria impact apoptosis will require the development of new tools with high spatial and temporal resolution. The development of such tools is beyond the scope of this manuscript.

Specific Comments

1. Mitochondrial fusion and fission events were visually quantified and authors claim that fission occurs significantly more often on the posterior side. Did the authors observe increased Drp1 localization on mitochondria in the posterior region? To strengthen the claim authors should Quantify Drp1 localization on mitochondria in the posterior vs. anterior sides.

Segos et al

In response to the reviewer's comment, we quantified the number of DRP-1 foci in the posterior and anterior halves of QL.p and in its daughter cells in real time. Since currently available N- or C-terminally tagged DRP-1 fusion proteins induce *drp-1* loss-of-function phenotypes (i.e. hyperfused mitochondria and partially penetrant embryonic lethality), we had to develop a new DRP-1 fusion protein. To that end, we edited the endogenous *C. elegans drp-1* locus using CRISPR/Cas-mediated genome editing and internally tagged it with mNeonGreen. Importantly, unlike C- or N-terminally tagged DRP-1 fusions, the internally tagged DRP-1 fusion (DRP-1(dx230Internal)) essentially behaves like the wild-type protein. Furthermore, using DRP-1(dx230Internal) and mitochondrial labelling with TMRE, we confirmed that DRP-1 foci essentially exclusively localise to mitochondria (see Lambie and Conradt, 2025).

Using DRP-1(dx230Internal), we identified DRP-1 foci in QL.p at metaphase and in QL.pa and QL.pp post-cytokinesis and determined the A/P ratio of the number of DRP-1 foci in QL.p and its daughter cells (see new paragraph in **Results page 5** and **Fig S5, new Parts h-j**). This revealed that DRP-1 foci are equally distributed among mitochondria in QL.p at metaphase. In contrast, post-cytokinesis, there are about 1.72-fold more DRP-1 foci per mitochondrial volume in QL.pp compared to QL.pa. This magnitude is consistent with our finding that mitochondrial fission events are 1.52-fold more frequent in the posterior. Based on this we suggest that localized mitochondrial fission may be caused by localized DRP-1 foci assembly.

2. Authors state: "our findings are consistent with the notion that unequal mitochondrial segregation during QL.p division and, hence mitochondrial density in QL.pp, correlates with the acquisition of the cell death fate by QL.pp."

As the evidence is correlative, the authors are to be applauded for arguing on this notion very carefully. However, the manuscript would greatly benefit from additional mechanistic insights. The key would be to address whether the asymmetric apportioning of mitochondria actually predisposes the cells to die. Do the mitochondria in QL.pp cells exhibit for example markers of damage?

Segos et al

As stated in our response to Reviewer 1, one of our future goals is to uncover the mechanisms through which mitochondria impact apoptosis in QL.pp. However, this is beyond the scope of the current study.

However, as outlined in our response to Reviewer 1, we have now looked at mitochondrial ROS (as an indicator of OXPHOS impairment), using a genetically encoded sensor for H₂O₂, HyPer7. Using this sensor, we determined mitochondrial ROS levels in QL.p and its daughter cells QL.pa and QL.pp. Our results demonstrate that there is no difference in ROS levels between mitochondria in QL.pa and QL.pp (see new paragraph "**Unequal mitochondrial segregation during QL.p division is not determined by mitochondrial ROS**" in **Results (page 8)** and new **Figure S14**). Therefore, ROS levels are unlikely to play a role in determining unequal mitochondrial segregation and in priming mitochondria in QL.pp for apoptosis.

3. It is unclear whether disrupting mitochondrial asymmetry in the mutant conditions affects

the fate acquisition of the daughter cells. Is the acquisition of the cell death fate by QL.pp also disrupted?

Segos et al

As shown in **Figures 4 and 5** and as described in the paragraph **Unequal mitochondrial segregation during QL.p division correlates with the acquisition of the cell death fate by QL.pp (pages 8-10)**, disrupting asymmetric mitochondrial segregation during QL.p division by introducing a *drp-1* or *fzo-1* loss-of-function mutation impacts the ability of QL.pp to adopt the cell death fate. For these experiments we combined super-resolution imaging with long-term cell fate tracking, using a microfluidic device. Using this experimental set up, we observed that QL.pp cells that had inherited relatively more mitochondria, took longer to die or even failed to die. Conversely, QL.pp cells that had inherited relatively fewer mitochondria died faster.

4. Mitochondria are classified into distinct classes based on their shape; however, the authors do not provide any data demonstrating that these mitochondria are functionally distinct. For example, they could examine the ultrastructure of these mitochondria, especially since they do not observe any differences in mitochondrial membrane potential (MMP).

Segos et al

As shown in **Figures 3 and S11** and as described in the paragraph **Unequal mitochondrial segregation during QL.p division is not determined by mitochondrial membrane potential (pages 6-7)**, we provide evidence that different volumetric classes of mitochondria are functionally distinct based on TMRE fluorescence intensities and that there is a correlation between size and shape on the one hand and TMRE fluorescence intensity on the other. Since we observe mitochondrial segregation in real time in developing *C. elegans* larvae, with currently available technology, we are unable to examine mitochondrial ultrastructure.

Minor comments:

“Post-cytokinesis, the two daughter cells generated were unequal in size 1 with QL.pa/QL.pp size ratios of 3.33 (lysosomes) and 3.25 (ER) (Fig. S1c, d, right).” The numbers presented in the text do not completely align with those shown in the figure.

Segos et al

We thank the review for pointing this out. We have corrected the sentence:

“Post-cytokinesis, the two daughter cells generated were unequal in size 1 with QL.pa/QL.pp size ratios of 3.25 (lysosomes) and 3.39 (ER) (**Fig. S1c, d, right**).”

The introduction provides an inaccurate portrayal of previous works and some key studies in the field are omitted. As a result, the authors claim for example that “Importantly, none of the studies in mammalian cells analysed mitochondria quantitatively and in real time as cells divide.” This is not true, and authors should also edit the text to reduce redundant statements. Moreover, some of the earlier work actually address mitochondria and other sub cellular compartments similarly to this study, and results should be discussed accordingly.

Segos et al

Based on the reviewer's comments, we have revised the **Introduction (Page 1, 2)**. We have made changes to the text, removed redundant statements (throughout the manuscript) and added additional references. In case specific references are still lacking, we would appreciate it, if the reviewer could point out the missing work. Concerning our statement "Importantly, none of the studies in mammalian cells analysed mitochondria quantitatively and in real time as cells divide.", what we meant with "analysing mitochondria quantitatively" is quantifying mitochondria through 3D rendering and taking cellular volume into account. We have clarified this now in the revised Introduction.

Reviewer #3 (Remarks to the Author):

It has been shown that asymmetric segregation of mitochondria during cell division determines the fate of daughter cells in aging yeast cells and mammalian stem-like cell lines. It is very likely that this is a more general mechanism, which also is important in metazoan development. Here, Segos et al. present the first in vivo evidence for this in living animals. Using *C. elegans* larvae they show that mitochondria segregate unequally during asymmetric division of a neuroblast that produces one daughter destined to differentiate and another daughter destined to die. The cell destined to die is much smaller and receives fewer mitochondria (also in relation to the cell volume), which are more fragmented. Interference with fusion/fission dynamics abrogates asymmetric inheritance of mitochondria. This correlates with the generation of some cells that inappropriately survive. Hence, the authors propose that non-random segregation of mitochondria contributes to cell fate divergence during asymmetric cell division. This is a beautiful manuscript. The hypotheses are well defined, the data are very clear and well controlled, the microscopy is impressive, and the manuscript is well written. I have only a few points that should be considered by the authors.

Segos et al

We thank the reviewer for their very positive feedback on the manuscript.

Important points:

1. Figure 2 shows that mitochondria are more fragmented in QL.pp than in QL.pa. It would be important to know whether this morphological difference emerges already before or only after cytokinesis. In other words: What is the evidence that mitochondrial fragmentation in the posterior half of QL.p starts already before the cytosols of the two daughter cells are separated? Figure S5 quantifies fusion/fission events from metaphase to post-cytokinesis. Can these data be shown separately for pre and post-cytokinetic cells?

Segos et al

The following observations and data indicate that the morphological differences between mitochondria in QL.pa and QL.pp emerge before the completion of cytokinesis:

(1) In Figure 2a, fission events (indicated by white arrowheads) occur at 3, 4, 5 and 6 min, which are all prior to the completion of cytokinesis (7 min, post-cytokinesis);

(2) as suggested by the reviewer, in Figure S5, we now separately show fusion/fission events during metaphase, anaphase and cytokinesis 1 and 2 (**Fig. S5, new Parts c-e**). This demonstrates that fission events occur during all stages, with an increased occurrence during anaphase and cytokinesis 1;

(3) using the microfluidic system, we have now determined the number of mitochondria in QL.pp post-cytokinesis until QL.pp death (**Fig. S5, new Parts f and g**). This demonstrates that the number of mitochondria is essentially constant during the length of the observation (up to 160 min). This suggests that there is essentially no mitochondrial fission (or fusion) occurring post-cytokinesis.

2. The Conradt lab has reported previously that block of DRP-1 activity causes inappropriate survival of cells during embryonic development by blocking apoptosis. How can the authors exclude that inappropriate survival of QL.pp cells in *drp-1* (lf) animals is exclusively due to interference with the apoptotic machinery rather than abnormal segregation of mitochondria? This should be discussed more extensively.

Segos et al

We thank the reviewer for this comment. Indeed, our results suggest that *drp-1* – but not *fzo-1* – impacts QL.pp survival not only through its contribution to unequal mitochondrial segregation (and, hence, mitochondrial density in QL.pp), but through its contribution to a second, mitochondrial segregation-independent process: As stated in **Results (page 10)**, once mitochondrial density is accounted for in the Cox regression model, the loss of *fzo-1* no longer shows a significant effect on QL.pp survival ($p = 0.065$). This suggests that the loss of *fzo-1* affects QL.pp survival exclusively through its contribution to unequal mitochondrial segregation. In contrast, the loss of *drp-1* maintains a significant effect on survival ($p = 0.005$), which suggests that the effect of the loss of *drp-1* on QL.pp survival is not solely due to *drp-1*'s contribution to unequal mitochondrial segregation. In the **Discussion (page 13)**, we now state that this second process may be related to the ability of DRP-1 protein to physically interact with a CED-9 Bcl-2/EGL-1 BH3-only complex:

*“Finally, our data indicate that in contrast to *fzo-1*, *drp-1*'s impact on QL.pp survival is not limited to its role in unequal mitochondrial segregation during QL.p division and, hence, its role in mitochondrial fission. Interestingly, we previously showed that DRP-1 protein - but not FZO-1 protein - can interact with CED-9 Bcl-2 bound to the pro-apoptotic BH3-only protein EGL-1⁴⁵. Therefore, we posit that *drp-1* may impact QL.pp survival also through a DRP-1/CED-9/EGL-1 complex, which may directly affect the activation or activity of the apoptosis machinery in QL.pp. Through what mechanism the DRP-1/CED-9/EGL-1 complex may affect apoptosis remains to be determined.”*

Minor points:

3. Figure S1: Why did the authors choose different ways of displaying their data in Figure S1 (lines versus dots)?

Segos et al

The reason why we displayed the data on lysosome segregation in a paired way (Fig S1C) but the data on ER segregation in an unpaired way (Fig S1D) is a difference in the transgenes used (lysosomes – eYFP::CUP-5, bcls159; ER – SP12::mCherry::KDEL, bcls160). The fluorescence intensity of bcls159 allowed us to image cells at two timepoints (metaphase and post-cytokinesis), which enabled us to display the data in a paired way. However, the fluorescence intensity of bcls160 was lower, and this made it impossible for us to image cells at two time points. Therefore, the data shown in Fig S1D for metaphase and post-cytokinesis were derived from different cells, which is the reason why it is displayed in an unpaired manner.

4. As fission and fusion must be balanced to maintain tubular mitochondria, it would make sense to indicate the fusion/fission ratios in Figure S5 in addition to the absolute numbers.

Segos et al

We thank the reviewer for the suggestion. We added the following statement in **Results (page 4)**:

“The ratio of fission to fusion events in the anterior is 1.48 whereas it is 5.00 in the posterior. This is consistent with balanced mitochondrial fission and fusion and the maintenance of mitochondrial morphology in the anterior but more mitochondrial fission and, hence, mitochondrial fragmentation in the posterior.”

5. Cell volume ratios should be added for the pig-1 mutant, which is supposed to divide symmetrically.

Segos et al

We now added data about cell volume ratios in pig-1 mutants (see **Fig. S6, new Part A**).

6. The order of the panels in the figures should be the same as the order they are mentioned in the text (for example Fig. 3b is mentioned in the text only after Fig. 3e).

Segos et al

This has been corrected:

*“TMRE staining was combined with nanobead-based immobilization and 3D rendering of cell and mitochondrial volumes to measure mitochondrial membrane potential (i.e. TMRE fluorescence intensity) within individual organelles (see Methods) (**Fig. 3a, b**).”*

7. Figure 3e: fzo-1 mutants should have fragmented mitochondria and drp-1 mutants should have hyperfused mitochondria. Why are some class 6 mitochondria found in fzo-1 and class 1 mitochondria in drp-1?

Segos et al

We observe mitochondria of different sizes in animals of all three genotypes; however, the number of organelles that are in a certain volumetric class (classes 1-6) changes from genotype

to genotype. For example, compared to wild type, there are more class 1 organelles in *fzo-1(lf)* and less in *drp-1(lf)* at both metaphase and post-cytokinesis (Fig 3E). In addition, differences in mitochondrial morphology between the three genotypes can be observed in the time series provided (Fig. 2, S4, S7, S8). There is evidence for Drp1-independent mitochondrial fission in mammals (PMID: 27181353). Therefore, small organelles in *drp-1(lf)* may be a result of drp-1-independent mitochondrial fission. Larger organelles in *fzo-1(lf)* are likely to arise from mitochondrial biogenesis i.e. growth. “Larger organelles” in *fzo-1(lf)* can also arise due to the physical proximity of two small organelles, which makes it challenging to resolve them as distinct organelles. This is a limitation of confocal microscopy with super-resolution implementation.

8. The authors should briefly explain in the text what a multivariate Cox proportional hazards regression is (first mentioned in line 348).

Segos et al

In response to the reviewer’s comment, we have modified the relevant paragraph in **Results (page 8, 9)**:

*“In order to be able to include the cases of inappropriately surviving QL.pp cells in *fzo-1(tm1133)* and *drp-1(bc455)* animals in our analyses, we used the Kaplan-Meier estimate. QL.pp survival probabilities of the different genotypes are shown in the Kaplan-Meier survival probability plot (Fig. 5a). The loss of *fzo-1* or *drp-1* shows a significant effect on QL.pp survival probability compared to wild type (+/+) (Fig. 5a; $p=0.0044$). In wild-type animals, essentially all QL.pp cells die within 300 min post-cytokinesis. In contrast, in *fzo-1(tm1133)*, some QL.pp cells die only after 900 min, and in *drp-1(tm1108)*, some survive longer than 1200 min. While a non-parametric and univariate Kaplan-Meier analysis is robust, it provides little information beyond the effect of genotype. In survival analysis, the multivariate Cox regression can be used when there are multiple potentially interacting covariates, such as for instance genotype and mitochondrial density. Using this proportional hazard model, we can relate the time before QL.pp death occurs (QL.pp survival time) to our covariates through the hazard function. This allows us to estimate the effect of mitochondrial density, while still accounting for the confounding effect of genotype in our time-to-event survival data. Holding the effect of genotype constant, we found that a 1% percent increase in mitochondrial density increases QL.pp’s life expectancy by 17.8% ($p = 0.002$). Looking at the relative death rate (which is cumulative over the entire experiment) (Fig. 5b), we see that increasing QL.pp mitochondrial density from 2.8% (average for wild type) to 7.1% leads to a halving of the relative death rate (from 1.0 to 0.5) (Fig. 5b).”*